# Multiple Invertible and Partial-Equivariant Function for Latent Vector Transformation to Enhance Disentanglement in VAEs

## Abstract

Disentanglement learning is a core issue for understanding and reusing trained information in Variational AutoEncoder (VAE), and an effective inductive bias has been reported as a key factor. However, the actual implementation of such bias is still vague. In this paper, we propose a novel method, called *Multiple Invertible and partial-equivariant transformation* (MIPE-transformation), to inject inductive bias by 1) guaranteeing the invertibility of latent-to-latent vector transformation while preserving a certain portion of equivariance of input-to-latent vector transformation, called *Invertible and partial-equivariant transformation* (IPE-transformation), 2) extending the form of prior and posterior in VAE frameworks to an unrestricted form through a learnable conversion into an approximate exponential family, called *Exponential Family conversion* (EF-conversion), and 3) integrating multiple units of IPE-transformation and EF-conversion, and their training. In experiments on 3D Cars, 3D Shapes, and dSprites datasets, MIPE-transformation improves the disentanglement performance of state-of-the-art VAEs.

## 1 Introduction

Disentanglement learning which aims to learn more interpretable representations is broadly useful in artificial intelligence fields such as classification (Singla et al., 2021), zero-shot learning (Tenenbaum, 2018), and domain adaptation (Li et al., 2019; Zou et al., 2020). A disentangled representation is defined as a change in a single dimension, which corresponds to unique semantic information. Several studies have been conducted based on this framework.

A major model for enhancing disentanglement learning is the Variational AutoEncoder (VAE) (Kingma & Welling, 2013). Based on the VAE, unsupervised disentangled representation learning has been developed (Higgins et al., 2017; Chen et al., 2018; Kim & Mnih, 2018; Jeong & Song, 2019; Li et al., 2020) through the factorizable variations and control of uncorrelatedness of each dimension in representations. Moreover, VAE models that handle the shape of the prior as a Gaussian mixture (Dilokthanakul et al., 2016b) or von Mises-Fisher (Davidson et al., 2018) were also developed, but disentanglement is still incomplete. (0) A critical study shows that unsupervised disentanglement learning is impossible without inductive bias (Locatello et al., 2019).

Recently, such inductive bias has been introduced from various perspectives on the transformation of latent vector space. Mathieu et al. (2018) (1) demonstrated that regularizing the aggregate posterior, rather than the posterior fitted to a complex prior, influences disentanglement learning. Intel-VAE (Miao et al., 2022) (2) utilized a complex prior and applied an invertible transformation to map the space to another latent space for a complex prior, providing better data representation. Group theory based bias also significantly improves in disentanglement (Zhu et al., 2021; Yang et al., 2022), which follows the definition by Higgins et al. (2018), based on group theory (3) even though these works implicitly inject the inductive bias. These works show that *equivariant* transformation between the input and latent vector spaces has a key role in disentanglement.

Inspired by the above works, we propose a *Multiple Invertible and partial-equivariant transformation* (MIPE-transformation) method, which is simply insertable into VAEs. In this paper, we mainly focus on two main aspects, as outlined below:

**Limitation of the Gaussian Prior.** (4) To address the limitations of the Gaussian prior, we propose the IPE-transformation, which generates an uncertain form of latent vector distributions, and we introduce a training procedure to align them to be close to an exponential family, called *exponential family conversion* (EF-conversion). This conversion allows the uncertain distribution to function within the typical training framework of VAEs to convey sufficient dataset structure (Mathieu et al., 2018; Miao et al., 2022). We then assume that an encoder is *partial-equivariant* function, a property we refer to as the *encoder equivariance condition.*

**Preserving Partial-Equivaraince.** (5) However the L2L transformation disrupts partial-equivariant function, defined by Romero & Lohit (2022), without condition. To address this issue, we implemented the latent-to-latent (L2L) transformation via an exponential matrix to satisfy the conditions necessary for preserving partial-equivariant; specifically, the transformation must be 1) symmetric and 2) invertible. (6) Then we mathematically show that a symmetric matrix exponential is an appropriate transformation to be a partial-equivariant function.

Through experiments involving both quantitative and qualitative analysis, MIPE-transformation demonstrates significant improvements in disentangled representation learning across the 3D Cars, 3D Shapes, and dSprites tasks. Our main contributions are summarized as follows.

1. We propose using a symmetric matrix exponential as a latent-to-latent vector transformation function to induce inductive bias, leveraging its invertible and equivariant properties with mathematical analysis.

2. We provide a training procedure and loss functions for VAEs to learn an unknown latent vector distribution as an approximate exponential family.

3. We propose the novel MIPE-transformation architecture, which integrates multiple IPE-transformation and EF-conversion, making it widely applicable to state-of-the-art VAEs.

4. We empirically analyze the properties of MIPE-transformation and validate its effectiveness in disentanglement learning on benchmark datasets.

Table 1: Terms and Notations

| | | | |
|---|---|---|---|
| $\boldsymbol{z}$ | Latent vector from encoder | $\psi(\cdot)$ | Invertible function |
| $\hat{\boldsymbol{z}}_m$ | Transformed latent vector by $\psi_m(\cdot)$ | $\hat{\boldsymbol{\epsilon}}_m$ | Transformed prior samples by $\psi_m(\cdot)$ |
| $\boldsymbol{\theta}_{\hat{\boldsymbol{z}}_m}$ | Natural Parameter of posterior | $\boldsymbol{\theta}_{\hat{\boldsymbol{\epsilon}}_m}$ | Natural Parameter of prior |
| $T$ | Sufficient Statistics | $A$ | Log-Normalizer |
| $\boldsymbol{\nu}$ | Evidence | $D_{\mathrm{KL}}(\cdot||\cdot)$ | Kullback-Leibler divergence |
| $f_{\boldsymbol{x}}(\cdot)$ | Power Density Function | $M_n(\mathbb{R})$ | A set of $n \times n$ real matrix |
| $GL_n(\mathbb{R})$ | General Linear Group | $Sym_n(\mathbb{R})$ | A set of $n \times n$ symmetric real matrix |
| $E_M$ | $\{\mathbf{e}^M | M \in M_n(\mathbb{R})\}$ | $E_S$ | $\{\mathbf{e}^S | S \in Sym_n(\mathbb{R})\}$ |
| $G_S$ | $G_S : (\mathbf{e}^S, *)$ | $G_I$ | Group of input space for symmetries |
| $G_L$ | Group of latent space for symmetries | $J$ | $G_S \cap G_L$ |
| $\psi_M(\cdot)$ | $\psi_M(\cdot) \in M_n(\mathbb{R})$ | $\psi_{E_M}(\cdot)$ | $\psi_{E_M}(\cdot) \in E_M$ |
| $\psi_{E_S}(\cdot)$ | $\psi_{E_S}(\cdot) \in E_S$ | $\mathbf{0}$ | zero vector |
| $\mathbf{0}_{n,n}$ | n by n zero matrix | $\mathcal{X}$ | Input space |
| $\mathcal{Z}$ | Latent vector space | $\hat{\mathcal{Z}}$ | Transformed latent vector space |
| $\Xi$ | $G_I \times G_L \rightarrow G_L$ | $\Gamma$ | $G_L \times G_T \rightarrow G_T$ |
| $\Xi^J$ | $G_I^J \times G_L^J \rightarrow G_L^J$ | $\Gamma^J$ | $G_L^J \times G_T^J \rightarrow G_T^J$ |

## 2 Related Work

Recently, various studies have focused on unsupervised disentanglement learning. Previous works are based on the definition by Bengio et al. (2013). One of the branches is InfoGAN (Chen et al., 2016) based works such as IB-GAN (Jeon et al., 2021), which implement an additional regularizer to improve informativeness (Eastwood & Williams, 2018). The other branch is based on the VAE. $\beta$-VAE (Higgins et al., 2017) penalizes the Kullback-Leibler divergence (KL divergence) using weighted hyper-parameters. Factor VAE (Kim & Mnih, 2018) and $\beta$-TCVAE (Chen et al., 2018) are trained using total correlation (TC) to encourage independent dimensions in a latent vector, employing a discriminator and decomposed components of the KL divergence term. Differently, we consider the recent disentanglement definition based on group theory (Higgins et al., 2018).

Following the definitions of disentangled representation learning based on group theory, several works have emphasized equivariant and improved disentangled representation learning. Commutative Lie Group VAE (CLG-VAE) (Zhu et al., 2021) proposed a direct mapping of the latent vector into Lie algebra to obtain a group structure (inductive bias) with constraints: commutative and hessian loss. Furthermore, Groupified VAE (Yang et al., 2022) utilizes the Spatial Broadcast Decoder (Watters et al., 2019) to implement an equivariant function to the cyclic group while guaranteeing the commutativity and invertibility of group actions. Topographic VAE (Keller & Welling, 2021) combines Student's-t distributions with variational inference and enforces rotated latent vectors to be equivariant. On the other hand, we apply an unrestricted prior and posterior for disentanglement learning.

There are several inductive biases to learning unsupervised disentanglement, such as group theory based and sequential order. In this section, we briefly discuss sequential order inductive bias even though its method is considered in different domains such as text and video frames. To individualize the static (time-invariant) and dynamic (time-variant), Li & Mandt (2018); Bai et al. (2021) proposed the latent variables one ($f$) is only dependent on the given times series datasets $x_{1:T}$, and the other ($\mathbf{z}_{1:T}$) is dependent on the $x_{1:T}$ and $f$. Moreover Bai et al. (2021) propose the novel ELBO with maximizing mutual information between the input and the latent vectors. These works empirically show that sequential order which includes separated latent vectors improves unsupervised disentanglement learning with diverse qualitative analysis. Differently in group theory based approaches, the proposed methods consider equivariant function between input and latent vector space.

Other VAE approaches implement other prior from Gaussian distribution to transformed Gaussian distribution, Gaussian mixture distribution (Dilokthanakul et al., 2016a) or von Mises-Fisher distribution (Davidson et al., 2018). Mathieu et al. (2018) ⑦ shows that model regularization with a Gaussian prior is inappropriate to convey sufficient dataset information and for disentanglement learning. Rather than using a Gaussian distribution, we employed an exponential family with high flexibility that encompasses a range of different distributions. ⑧ InteL-VAE (Miao et al., 2022) utilize the invertible function to transform a Gaussian distribution to a complex distribution as Mathieu et al. (2018). Differently, we show a clearer relation of invertibility to disentanglement and improve VAEs to use its unrestricted form of prior.

Invertible and equivariant Deep Neural Networks have been investigated with normalizing flows. As proven by Xiao & Liu (2020), utilized matrix exponential on Neural networks is invertible, but it only provides mathematical foundations of the transformation. Matrix exponential is utilized to implement an invertible and equivariant function to improve the generative flow compare to linear function Hoogeboom et al. (2020). To specify the exponential family, other works contribute uncertainty of exponential family distribution with Bayesian update Charpentier et al. (2020; 2022). In addition, Ranganath et al. (2014) hierarchically controls the natural parameter across the layers and determines the exponential family distribution with the moment of sufficient statistic. In our work, we show how to use it for disentanglement learning.

## 3 Method

The overview of a VAE equipped with MIPE-transformation is shown in Figure 1. ⑨ We aim to explain the three core components of the MIPE-transformation. In Section 3.1, we mathematically prove why the

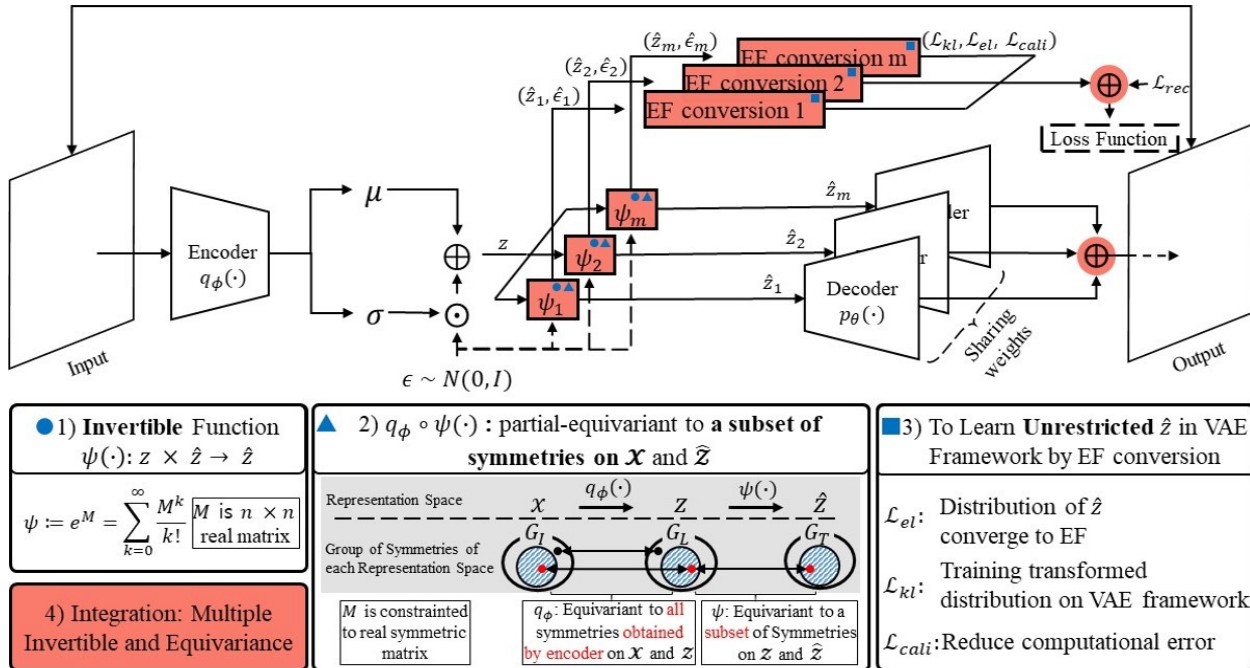

Figure 1: The overall architecture of our proposed *MIPET*-VAE. The invertible and partial-equivariant function $\psi(\cdot)$ for L2L transformation consists of a symmetric matrix exponential to be 1) invertible and 2) partial-equivariant. Then 3) EF conversion module converges the distribution of unrestricted $\hat{z}$ to be EF with $\mathcal{L}_{el}$ loss. Also, it applies KL divergence loss ($\mathcal{L}_{kl}$) between the transformed posterior and prior, which are expressed by the power density function of EF. In the last, EF conversion reduces the computational error ($\mathcal{L}_{cali}$) between approximated and true KL divergence. 4) The reddish color represents the integration parts. The blue figures represent each property. The details of the gray box are in Figure 2.

*IPE-transformation*, which transforms latent vectors, better preserves partial-equivariance when it is an invertible and symmetric matrix. In Section 3.2, we describe *EF-conversion*, which extends the Gaussian distribution to a diverse exponential family distribution. Finally, in Section 3.3, we illustrate how multiple IPE-transformations and EF-conversions can be integrated into a VAE-based model to implicitly inject inductive bias.

## 3.1 Partial-Equivariant Function and Invertible L2L Transformation

### 3.1.1 Why Should L2L Transformation Be Equivariant?

Let's consider equivariant function between the input and transformed latent vector space, directly used for a decoder in the VAE frameworks. All L2L transformations do not extend the encoder equivariance condition to the relation between input and transformed latent space. This problem is more precisely shown in Figure 2, which illustrates partial equivariance condition over the input space $\mathcal{X}$, latent vector space $\mathcal{Z}$, and its transformed latent vector space $\hat{\mathcal{Z}}$ with a corresponding group of symmetries $G_I$, $G_L$, and $G_T$, respectively. In the VAEs literature, it has not been reported to restrict L2L transformation to guarantee equivariant function between two spaces, so we propose a solution to guarantee at least a part of symmetries to be equivariant.

### 3.1.2 Equivariance Property with Symmetric Matrix Exponential

To enhance the equivariance of L2L transformation, ⑩ we focus on which transformation is appropriate. Then we prove that an invertible and symmetric matrix preserves partial-equivariance better than other

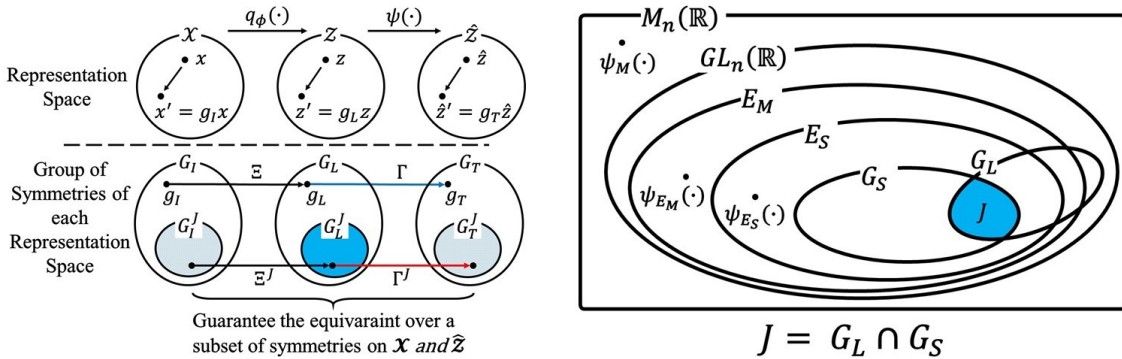

Figure 2: $G_I$, and $G_L$ are obtained through encoder $q_\phi$ (encoder equivariance condition). The left side figure shows the relation between each space and symmetries. If $\psi(\cdot)$ is equivariant function over all $G_L$, and $G_T$, then there exist $\Gamma$, where $\Gamma : G_L \to G_T$, and $\Xi \circ \Gamma : G_I \to G_T$. However, unrestricted $\psi(\cdot)$ has no guarantee to be partial- or full-equivariant. The red arrows represent our method: L2L transformation guarantees $\Gamma^J : G_L^J \to G_T^J$, and $\Xi^J \circ \Gamma^J : G_I^J \to G_T^J$, given the encoder equivariance condition $\Xi : G_I \to G_L$.

matrices. We show that 1) a group with symmetric and invertible matrices guarantees equivariance of $\psi(\cdot)$ over the specific group $G_S$, 2) this $\psi(\cdot)$ being equivariant over subset of symmetries between the input space and transformed latent vector space (11) to show the connection between two spaces, and 3) (12) the invertible and symmetric matrix (symmetric matrix exponential) increases the probability of $\psi(\cdot)$ to be in the group (equal to be equivariant over the subset of symmetries).

We particularly call the transformations as *symmetries* (Higgins et al., 2022) to distinguish them from IPE- and I2L-transformations. For the generality of our method, we consider an arbitrary VAE model that has no restriction on creating intersections to any set as Figure 2.

**Proposition 3.1.** *Any $\psi(\cdot) \in G_S$, notated as $\psi_{G_S}(\cdot)$, is equivariant to group $G_S$.*

*Proof.* The group $G_S$ is closed to matrix multiplication, and its element is always a symmetric matrix by definition. Then, any two elements in $G_S$ are commutative because if matrix multiplication of two symmetric matrices is symmetric then both are commutative. As a result, $\psi_{G_S}(\cdot)$ and group elements of $G_S$ are commutative ($G_S$ is an abelian group). Because of the commutativity, $\psi_{G_S}(g_s \circ z) = \mathbf{e}^S g_s z = g_s \mathbf{e}^S z = g_s \circ \psi_{G_S}(z)$ for $g_s \in G_S$ if the group action $\circ$ is set to matrix multiplication, where $\psi_{G_S} \in G_S$. This equation satisfies the general definition of an equivariant function that a function $f(\cdot)$ is equivariant if $f(g \circ z) = g \circ f(z)$ for all $g$ in a group $G$ by matching $f$, $g$, and $G$ to $\psi_{G_S}$, $g_s$, and $G_S$, respectively. ∎

**Proposition 3.2.** *If $q_\phi$ is equivariant over defined on group of symmetries $G_I^J$ and $G_L^J$, then $\psi_{G_S}(q_\phi(\cdot))$ is equivariant to symmetries in $G_I$ corresponding to $G_S \cap G_L$ and $G_T$ corresponding to $G_S \cap G_L$ by the equivariance of $q_\phi$.*

*Proof.* The function $\psi_{G_S}(\cdot)$ is an equivariant function over group elements in $G_S \cap G_L$ by Proposition 3.1. Then, the composite function, $\psi_{G_S}(\cdot)$ and $q_\phi$, is an equivariant function of $G_I$ corresponding to $G_S \cap G_L$ and $G_T$ corresponding to $G_S \cap G_L$. Let $g_L^J$ be a group element in $G_S \cap G_L$, and $g_I^J$ is a group element in $G_I$ corresponding to $G_S \cap G_L$. More details are in Appendix C.1. ∎

Therefore, the invertible and symmetric matrix is a partial-equivariant function between the data space and transformed latent vector space.

**Proposition 3.3.** $Pr(\psi_{E_S}(\cdot) \in G_S) > Pr(\psi_{E_M}(\cdot) \in G_S) > Pr(\psi_M(\cdot) \in G_S)$.

*Proof.* All $\mathbf{e}^S \in E_S$ are in $E_M$ since $Sym_n(\mathbb{R}) \subset M_n(\mathbb{R})$. However, $E_M \not\subset E_S$ because $\mathbf{e}^S$ is always symmetric, but $\mathbf{e}^M$ can be an asymmetric matrix. Therefore $E_M \not\subset E_S$. Therefore, the probability $Pr(\psi_{E_S}(\cdot) \in G_S) = \frac{P(G_S)}{P(E_S)}$ is greater than $Pr(\psi_{E_M}(\cdot) \in G_S) = \frac{P(G_S)}{P(E_M)}$. In the same way, $Pr(\psi_{E_M}(\cdot) \in G_S) > Pr(\psi_M(\cdot) \in G_S) = \frac{P(G_S)}{P(M_n(\mathbb{R}))}$ because $E_M \subset M_n(\mathbb{R})$ and non-invertible functions are only in $M_n(\mathbb{R})$. ∎

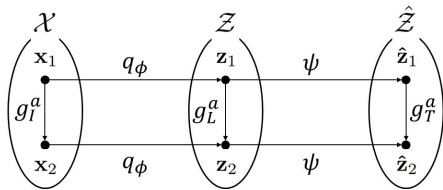

Figure 3: Equivariant map: $\mathcal{X}$, $\mathcal{Z}$, and $\hat{\mathcal{Z}}$ are input space, latent vector space, and transformed latent vector space by L2L transfomration function $\psi(\cdot) : \mathbb{R}^n \to \mathbb{R}^n$. respectively. $\boldsymbol{x} \in \mathcal{X}$, $\boldsymbol{z} \in \mathcal{Z}$, and $\hat{\boldsymbol{z}} \in \hat{\mathcal{Z}}$.

Therefore, $\psi_{E_S}$ clearly increases the probability of preserving a certain type of equivariance compared to unrestricted $\psi$ functions. (13) It implies that symmetric and invertible properties for a matrix are necessary to preserve partial equivariance.

The conditional probability $Pr(\psi_{E_S}(\cdot) \in G_S)$, $Pr(\psi_{E_M}(\cdot) \in G_S)$, and $Pr(\psi_M(\cdot) \in G_S)$ is changed by the distribution of the observation of $\psi(\cdot)$, which depends on the model parameters. However, the inequality $Pr(\psi_{E_S}(\cdot) \in G_S) > Pr(\psi_{E_M}(\cdot) \in G_S) > Pr(\psi_M(\cdot) \in G_S)$ is not changed regardless of the distribution of observation of $\psi(\cdot)$. We empirically validate the impact of equivariance with the uncertain $P(\cdot)$ to disentanglement in Section D.4.

This correspondence of decomposition is expected to transfer the independence between dimensions of $\boldsymbol{z}$ to the space of $\hat{\boldsymbol{z}}$ (Higgins et al., 2018).

### 3.1.3 Invertible Property by Using Matrix Exponential

To guarantee the invertible property of IPE-transformation, we use a function $\psi(\cdot) = \mathbf{e}^{\boldsymbol{M}} * \cdot$ for the transformation, where $\boldsymbol{M}$ is in $n \times n$ real number matrix set $M_n(\mathbb{R})$ (Xiao & Liu, 2020). The operator $*$ is matrix multiplication, and $\mathbf{e}^{\boldsymbol{M}} = \sum_k^\infty \frac{\boldsymbol{M}^k}{k!}$. Our motivation is to use the benefits of injecting explicit inductive bias for disentanglement (Locatello et al., 2019; Miao et al., 2022). InteL-VAE effectively extracts hierarchical representation, which includes low-level features (affect to a specific factor) and high-level features (affect to complex factors) with an invertible transformation function (Miao et al., 2022).

### 3.2 Exponential Family Conversion for Unknown Prior

In VAE frameworks, the Gaussian normal distribution is applied as a prior. However, a prior from data is usually unknown and may not follow the Gaussian distribution (Miao et al., 2022). As a solution, we present a training procedure for VAEs to build an exponential family distribution from a latent variable of an arbitrary distribution. Then, we introduce training losses obtained from the unit IPE-transformation function and EF-conversion.

### 3.2.1 Elements of Exponential Family Distribution Settings

First, we set sufficient statistics $T(\cdot)$, log-normalizer $A(\cdot)$, and carrier or base measure $B(\cdot)$ are deterministic functions by maximizing conjugate prior for parameter $\xi$. To determine the *natural parameter* of posterior and prior $\boldsymbol{\theta}_{\hat{\boldsymbol{z}}_m}$, and $\hat{\boldsymbol{\epsilon}}_m$, we use a natural parameter generator (NPG) designed by multi-layer perceptron (Charpentier et al., 2022). As introduced in Bishop (2006); Charpentier et al. (2022), we assume exponential family always admits a conjugate prior:

$$q(\boldsymbol{\theta}|\xi, \boldsymbol{\nu}) = \exp(\boldsymbol{\nu}\boldsymbol{\theta}^\mathsf{T}\xi - \boldsymbol{\nu}A(\boldsymbol{\theta}) + B'(\xi, \boldsymbol{\nu})), \tag{1}$$

where $B'(\cdot)$ is a *normalize coefficient* and $\nu$ is evidence, and it is expressed by prior natural parameter $\xi$. However, generated natural parameter $\boldsymbol{\theta}_{\hat{\boldsymbol{z}}_m}$ is not guaranteed as the appropriate parameter of the exponential family corresponds to conjugate prior. To satisfy this condition, we assume observation is a set of independent identically distributed, then Eq. 13 is modified: $p(\mathbf{X}|\boldsymbol{\theta}) = \prod_{n=1}^N h(\mathbf{x}_n)\exp(\boldsymbol{\theta}^\mathsf{T}\sum_{n=1}^N T(\mathbf{x}_n) - A(\boldsymbol{\theta}))$ (Bishop,

2006), where observation $\mathbf{X} = \{\mathbf{x}_1, \cdots \mathbf{x}_N\}$. In the next, we multiply the modified formation by the prior Eq. 1 to obtain the posterior distribution (Bishop, 2006) as Eq. 2.

### 3.2.2 Distribution Approximation As an Exponential Family

The procedure represents a posterior distribution in the exponential family by adopting the following form:

$$p(\boldsymbol{\theta}|\mathbf{X}, \xi, \boldsymbol{\nu}) \propto \exp(\boldsymbol{\theta}^{\mathsf{T}}(\sum_{n=1}^{N} T(\mathbf{x}_n) + \boldsymbol{\nu}\xi) - A(\boldsymbol{\theta})), \tag{2}$$

where *sufficient statistics* $T(\cdot)$ and *log-normalizer*, $A(\cdot)$ are known functions, samples $\mathbf{X} = \{\mathbf{x}_1, \mathbf{x}_2, \ldots, \mathbf{x}_n\}$ from distribution, and *natural parameter* of posterior $\boldsymbol{\theta}$ and of prior $\xi$ (Bishop, 2006). The functions $T(\cdot)$, and $A(\cdot)$ are deterministic functions to maximize posterior distribution. The *evidence* is implemented as learnable parameters $\boldsymbol{\nu} \in \mathbb{R}^{n \times n}$. The natural parameter is generated by a multi-layer perceptron as Charpentier et al. (2022). This general form approximating an exponential family distribution with learnable parameters can extend VAEs to use a wider distribution for latent variables by simply matching $\mathbf{X}$ to generated latent variables. After IPE-transformation, we can apply the form by using the $\hat{\boldsymbol{z}}_m$, $\boldsymbol{\theta}_{\hat{\boldsymbol{z}}_m}$, and $\boldsymbol{\theta}_{\hat{\boldsymbol{\epsilon}}_m}$ for $\mathbf{X}$, $\boldsymbol{\theta}$, and $\xi$, respectively.

### 3.2.3 EF Similarity Loss

We added a loss to converge the unrestricted distributions of $\hat{\boldsymbol{z}}$ to the power density function of the exponential family by constraining the posterior maximization as:

$$\text{maximize } \log p(\boldsymbol{\theta}_{\hat{\boldsymbol{z}}_m}|\hat{\boldsymbol{z}}_m, \boldsymbol{\theta}_{\hat{\boldsymbol{\epsilon}}_m}, \boldsymbol{\nu}_m) \quad \text{s.t. } D_{\mathrm{KL}}(f_{\boldsymbol{x}}(\boldsymbol{x}|\boldsymbol{\theta}_{\hat{\boldsymbol{z}}_m})||f_{\boldsymbol{x}}(\boldsymbol{x}|\boldsymbol{\theta}_{\hat{\boldsymbol{\epsilon}}_m})) \geq 0 \tag{3}$$

$$\Rightarrow \mathcal{L}_s(\hat{\boldsymbol{z}}_m, \hat{\boldsymbol{\epsilon}}_m) = \log p(\boldsymbol{\theta}_{\hat{\boldsymbol{z}}_m}|\hat{\boldsymbol{z}}_m, \boldsymbol{\theta}_{\hat{\boldsymbol{\epsilon}}_m}, \boldsymbol{\nu}_m) + \boldsymbol{\lambda}_m D_{\mathrm{KL}}(f_{\boldsymbol{x}}(\boldsymbol{x}|\boldsymbol{\theta}_{\hat{\boldsymbol{z}}_m})||f_{\mathbf{x}}(\boldsymbol{x}|\boldsymbol{\theta}_{\hat{\boldsymbol{\epsilon}}_m})) \tag{4}$$

$$\Rightarrow \mathcal{L}_{el} := ||\nabla_{\hat{\boldsymbol{z}}_m, \hat{\boldsymbol{\epsilon}}_m, \boldsymbol{\lambda}_m} \mathcal{L}_s||_2^2 = 0. \tag{5}$$

The notation $\boldsymbol{\theta}_k$ is a generated natural parameter by a given $k \in \{\hat{\boldsymbol{z}}, \hat{\boldsymbol{\theta}}\}$, and $f_{\boldsymbol{x}}(\boldsymbol{x}|\boldsymbol{\theta})$ is a power density function of the exponential family. Moreover, $\boldsymbol{\lambda}_m$ is a trainable parameter for optimizing the Lagrange multiplier, and $D_{\mathrm{KL}}(f_{\boldsymbol{x}}(\boldsymbol{x}|\boldsymbol{\theta}_{\hat{\boldsymbol{z}}_m})||f_{\boldsymbol{x}}(\boldsymbol{x}|\boldsymbol{\theta}_{\hat{\boldsymbol{\epsilon}}_m}))$ is a KL divergence of the exponential family.

### 3.2.4 KL Divergence for Evidence of Lower Bound

The KL divergence of Gaussian distribution (Kingma & Welling, 2013) is computed using mean and variance, which are the parameters of a Gaussian distribution. To introduce a loss as the KL divergence of Gaussian distribution, we compute KL divergence of the exponential family in Eq. 2 using the learnable parameter $T(\cdot)$ and $A(\cdot)$ with given natural parameter $\boldsymbol{\theta}_{\hat{\boldsymbol{z}}}$ and $\boldsymbol{\theta}_{\hat{\boldsymbol{\epsilon}}}$, expressed as:

$$\begin{aligned} \mathcal{L}_{kl} &:= D_{\mathrm{KL}}(f_{\boldsymbol{x}}(\boldsymbol{x}|\boldsymbol{\theta}_{\hat{\boldsymbol{z}}_m})||f_{\boldsymbol{x}}(\boldsymbol{x}|\boldsymbol{\theta}_{\hat{\boldsymbol{\epsilon}}_m})) \\ &= A(\boldsymbol{\theta}_{\hat{\boldsymbol{\epsilon}}}) - A(\boldsymbol{\theta}_{\hat{\boldsymbol{z}}}) + \boldsymbol{\theta}_{\hat{\boldsymbol{z}}}^{\mathsf{T}} \nabla_{\boldsymbol{\theta}_{\hat{\boldsymbol{z}}}} A(\boldsymbol{\theta}_{\hat{\boldsymbol{z}}}) - \boldsymbol{\theta}_{\hat{\boldsymbol{\epsilon}}}^{\mathsf{T}} \nabla_{\boldsymbol{\theta}_{\hat{\boldsymbol{\epsilon}}}} A(\boldsymbol{\theta}_{\hat{\boldsymbol{\epsilon}}}). \end{aligned} \tag{6}$$

### 3.2.5 KL Divergence Calibration Loss

To reduce the error between the approximation and true matrix for the matrix exponential (Bader et al., 2019), we add a loss to minimize the difference of their KL divergence measured by mean squared error (MSE) as:

$$\mathcal{L}_{cali} = \mathrm{MSE}(D_{\mathrm{KL}}(q_\phi(\boldsymbol{z}|\boldsymbol{x})||p_\theta(\boldsymbol{z})), D_{\mathrm{KL}}(f_{\boldsymbol{x}}(\boldsymbol{x}|\boldsymbol{\theta}_{\hat{\boldsymbol{z}}_m})||f_{\boldsymbol{x}}(\boldsymbol{x}|\boldsymbol{\theta}_{\hat{\boldsymbol{\epsilon}}_m}))), \tag{7}$$

which is the KL divergence calibration loss ($\mathcal{L}_{cali}$).

### 3.2.6 Implicit Semantic Mask

We propose an implicit semantic mask to improve disentanglement learning. We apply mask matrix $\mathcal{M}$ which consists of 0 or 1 element to log-normalizer to prevent less effective weight flow as:

$$\mathcal{M}_{ij} = \begin{cases} 1 \text{ if } |\mathcal{W}_{ij}| \geq \mu_{|\mathcal{W}_{ij}|} - \lambda \sigma_{|\mathcal{W}_{ij}|} \\ 0 \text{ otherwise} \end{cases}, \tag{8}$$

where $\mathcal{W}$ is the weight of log-normalizer, $\lambda$ is a hyper-parameter, $\mu_{|\mathcal{W}_{ij}|}$, and $\sigma_{|\mathcal{W}_{ij}|}$ are the mean, and standard deviation of weight respectively. Previous work (Yang et al., 2020) utilizes a semantic mask in input space directly, but we inject the semantic mask implicitly on the latent space.

### 3.3 Integration for Multiple IPE-Transformation and EF-Conversion

We mathematically extend IPE-transformation to MIPE-transformation, which is the equivalent process of $\beta$-VAE to enhance disentanglement. Each IPE-transformation function operates independently, then the reconstruction error for objective function is defined as:

$$\mathcal{L}_{rec} \coloneqq \frac{1}{k} \sum_{i=1}^{k} \left[ \int q_i(\hat{z}_i|\boldsymbol{x}) \log p_\theta(\boldsymbol{x}|\hat{z}_i) \mathrm{d}\hat{z}_i \prod_{j=1, j\neq i}^{k} \int q_j(\hat{z}_j|\boldsymbol{x}) \mathrm{d}\hat{z}_j \right] = \frac{1}{k} \sum_{i=1}^{k} E_{q_{\phi,\psi_i}(\boldsymbol{z}|\boldsymbol{x})} \log p_\theta(\boldsymbol{x}|\psi_i(\boldsymbol{z})), \quad (9)$$

where $\hat{z}_i = \psi_i(\boldsymbol{z})$. Therefore, we define ELBO as:

$$\mathcal{L}'(\phi, \theta, \psi_{i\in[1,k]}; \boldsymbol{x}) = \frac{1}{k} \sum_{i=1}^{k} \mathbb{E}_{q_{\phi,\psi_i}(\boldsymbol{z}_i|\boldsymbol{x})} \log p_\theta(\boldsymbol{x}|\psi_i(\boldsymbol{z})) - \sum_{i=1}^{k} D_{\mathrm{KL}}(q_{\phi,\psi_i}(\boldsymbol{z}|\boldsymbol{x})||p_{\psi_i}(\boldsymbol{z})). \tag{10}$$

However, following Eq. 10, k samples are generated, and each sample is disentangled for different factors. We implement the output as the average of the sum of the k samples to obtain a single sample with a superposition effect from k samples. Moreover, the KL divergence term in Eq. 10 represents that increasing number of MIPE-transformation is equal to an increasing $\beta$ hyper-parameter in $\beta$-VAE (Higgins et al., 2017) and ⑭ more details are in Appendix C.1.

The VAEs equipped with MIPE-transformation (MIPET-VAEs) can be trained with the following loss:

$$\mathcal{L}(\phi, \theta, \psi_{i\in[1,k]}; \boldsymbol{x}) = \mathcal{L}_{rec} - \mathcal{L}_{kl} - \mathcal{L}_{el} - \mathcal{L}_{cali}. \tag{11}$$

## 4 Experiment Settings

### 4.0.1 Models

As baseline models, we select VAE (Kingma & Welling, 2013), $\beta$-VAE (Higgins et al., 2017), $\beta$-TCVAE (Chen et al., 2018), Factor-VAE (Kim & Mnih, 2018), Control-VAE (Shao et al., 2020),and CLG-VAE (Zhu et al., 2021). These models are compared to their extension to adopt MIPET, abbreviated by adding the MIPET prefix. We apply the proposed method to $\beta$-TCVAE only with the EF similarity loss term because $\beta$-TCVAE penalizes the divided KL divergence terms. We set the same encoder and decoder architecture in each model to exclude the overlapped effects. Also, we follow the same model architecture which are introduced in previous works (Kim & Mnih, 2018) and model details are in Table 7-8.

### 4.0.2 Datasets

We compare well-known VAEs to MIPET-VAEs on the following data sets with 1) dSprites (Matthey et al., 2017) which consists of 737,280 binary $64 \times 64$ images of dSprites with five independent ground truth factors(number of values), *i.e.* shape(3), orientation(40), scale(6), x-position(32), and y-position(32). 2) 3D Shapes (Burgess & Kim, 2018) which consists of 480,000 RGB $64 \times 64 \times 3$ images of 3D Shapes with six independent ground truth factors: shape(4) orientation(15), scale(8), wall color(10), floor color(10), and object color(10). 3) 3D Cars (Reed et al., 2015) which consists of 17,568 RGB $64 \times 64 \times 3$ images of 3D Shapes with three independent ground truth factors: car models(183), azimuth directions(24), and elevations(4).

Table 2: Performance (mean ± std) of four metrics on dSprites, 3D Shapes, and 3D Cars.

| dSprites | FVM ↑ | | MIG ↑ | | SAP ↑ | | DCI ↑ | |
|---|---|---|---|---|---|---|---|---|
| | original | MIPET | original | MIPET | original | MIPET | original | MIPET |
| $\beta$-VAE | 69.15(±5.88) | **74.19**(±5.62) | 9.49(±8.30) | **19.72**(±11.37) | 2.43(±2.07) | **5.08**(±2.90) | 18.57(±12.41) | **28.81**(±10.19) |
| $\beta$-TCVAE | 78.50(±7.93) | **79.87**(±5.80) | 26.00(±9.06) | **35.04**(±4.07) | 7.31(±0.61) | **7.70**(±1.63) | 41.80(±8.55) | **47.83**(±5.01) |
| Factor-VAE | 67.78(±7.48) | **68.38**(±8.55) | 14.67(±10.40) | **19.31**(±13.46) | 2.35(±2.32) | **4.24**(±3.36) | 22.58(±8.50) | **30.18**(±12.99) |
| CLG-VAE | 79.06(±6.83) | **81.80**(±3.17) | 23.40(±7.89) | **36.34**(±5.55) | 7.37(±0.96) | **8.03**(±0.83) | 37.68(±7.83) | **44.73**(±5.11) |
| Control-VAE | 62.36(±8.62) | **67.71**(±6.41) | 4.36(±2.86) | **7.34**(±4.10) | **2.11**(±1.88) | 1.93(±1.63) | 10.40(±3.42) | **15.18**(±4.61) |

| 3D Shapes | FVM ↑ | | MIG ↑ | | SAP ↑ | | DCI ↑ | |
|---|---|---|---|---|---|---|---|---|
| | original | MIPET | original | MIPET | original | MIPET | original | MIPET |
| $\beta$-VAE | 71.76(±12.26) | **75.19**(±8.16) | 37.33(±22.34) | **47.37**(±10.13) | 7.48(±4.12) | **9.20**(±2.44) | 52.07(±17.92) | **54.95**(±8.99) |
| $\beta$-TCVAE | 76.62(±10.23) | **80.59**(±8.57) | 52.93(±20.5) | **54.49**(±9.44) | 10.64(±5.93) | **11.58**(±3.32) | 65.32(±11.37) | **66.22**(±7.32) |
| Factor-VAE | 77.43(±10.71) | **78.34**(±8.35) | 24.23(±26.13) | **48.59**(±10.58) | 4.99(±4.46) | **9.84**(±2.73) | **68.68**(±12.92) | 60.23(±9.90) |
| CLG-VAE | 77.04(±8.22) | **80.17**(±8.43) | 49.74(±8.18) | **53.87**(±7.41) | 9.20(±2.44) | **12.83**(±3.01) | 57.70(±8.60) | **60.74**(±7.77) |
| Control-VAE | 71.05(±14.35) | **71.89**(±8.33) | 24.88(±13.68) | **32.28**(±10.74) | 6.60(±3.59) | **7.14**(±2.09) | 40.08(±13.45) | **43.06**(±8.68) |

| 3D Cars | FVM ↑ | | MIG ↑ | | SAP ↑ | | DCI ↑ | |
|---|---|---|---|---|---|---|---|---|
| | original | MIPET | original | MIPET | original | MIPET | original | MIPET |
| $\beta$-VAE | **89.48**(±5.22) | 88.95(±5.94) | 6.90(±2.70) | **7.27**(±1.99) | 1.30(±0.48) | **1.88**(±1.12) | **19.85**(±4.87) | 18.90(±4.49) |
| $\beta$-TCVAE | 95.84(±3.40) | **96.43**(±2.42) | **11.87**(±2.90) | 10.80(±1.22) | 1.55(±0.38) | **1.88**(±1.12) | **27.91**(±4.31) | 26.08(±2.47) |
| Factor-VAE | 89.64(±2.62) | **93.66**(±3.83) | **10.63**(±1.30) | 9.43(±1.12) | 2.51(±0.37) | **3.28**(±0.83) | **26.27**(±3.17) | 23.13(±0.72) |
| CLG-VAE | 86.11(±7.12) | **91.06**(±5.09) | 6.19(±2.42) | **8.51**(±2.11) | **2.06**(±0.60) | 1.99(±0.93) | 16.91(±4.01) | **18.31**(±2.83) |
| Control-VAE | 88.76(±7.66) | **89.10**(±6.90) | 4.68(±2.67) | **5.08**(±2.68) | 1.16(±0.74) | **1.45**(±0.86) | 14.70(±3.84) | **15.22**(±4.15) |

Table 3: $p$-value of t-test for original vs MIPET results of Table 2, which are averaged over models (bold: positive and significant, italic: positive but insignificant, normal: lower performance).

| $p$-value | VAEs | | | | CLG-VAE | | | | $\beta$-TCVAEs | | | |
|---|---|---|---|---|---|---|---|---|---|---|---|---|
| | FVM | MIG | SAP | DCI | FVM | MIG | SAP | DCI | FVM | MIG | SAP | DCI |
| dSprites | **0.000** | **0.000** | **0.000** | **0.000** | **0.030** | **0.000** | **0.005** | **0.000** | *0.281* | **0.000** | *0.170* | **0.009** |
| 3D Shapes | **0.080** | **0.007** | **0.016** | *0.191* | *0.085* | **0.029** | **0.000** | *0.088* | *0.111* | *0.383* | *0.277* | *0.390* |
| 3D Cars | 0.659 | *0.250* | **0.003** | 0.583 | **0.003** | **0.000** | 0.630 | *0.071* | *0.278* | 0.923 | *0.119* | 0.933 |

### 4.0.3 Training

We set 256 mini-batch size in the datasets (dSprites, 3D Shapes, and 3D Cars), Adam optimizer with learning rate $4 \times 10^{-4}$, $\beta_1 = 0.9$, $\beta_2 = 0.999$, and epochs from $\{30, 67, 200\}$ as a common setting for all the comparative methods. For the comparison, we follow training and inference on the whole dataset. We train each model for 30, 67, and 200 epochs on the dSprites, 3D Shapes, and 3D Cars, respectively, as introduced in Kim & Mnih (2018); Ren et al. (2022). We tune $\beta$ from $\{1, 2, 4, 10\}$ and $\{4, 6\}$ for $\beta$-VAE and $\beta$-TCVAE, respectively. The $\alpha = 1$ and $\gamma = 1$ of $\beta$-TCVAE as Chen et al. (2018). We set the dimension size of the latent vectors from $\{6, 10\}$ for 10 on dSprites and 3D Cars datasets and 6 for 3D Shapes, but we set 10 for CLG-VAE because it sets 10 dimensions size on 3D Shapes in Zhu et al. (2021). Regarding the CLG-VAE, we fix $\lambda_{\text{decomp}}$, $\lambda_{\text{hessian}}$, and forward group features as 40, 20, and 0.2, respectively. Because the hyper-parameters showed the best result in Zhu et al. (2021). We set group reconstruction from $\{0.2, 0.5, 0.7\}$. For Control-VAE, we set the maximum KL divergence value from $\{10, 12, \ldots, 20\}$. In addition, we set masking ratio $\lambda$ from $\{0.0, 0.5, \ldots 2.0, \infty\}$. To check the impact of MIPE-transformation, we do not consider the Groupified VAE because the latter is implemented with an extended decoder (different capacity).

### 4.0.4 Evaluation

We conduct experiments on NVIDIA A100, RTX 2080 Ti, and RTX 3090. We set 100 samples to evaluate global empirical variance in each dimension and run it a total of 800 times to estimate the FVM score introduced in Kim & Mnih (2018). For the MIG (Chen et al., 2018), SAP (Kumar et al., 2018), and DCI (Eastwood & Williams, 2018), we follow default values introduced in Michlo (2021), training and evaluation 100 and 50 times with 100 mini-batches, respectively. We evaluate four disentanglement metrics for a less biased understanding of the actual states of disentanglement.

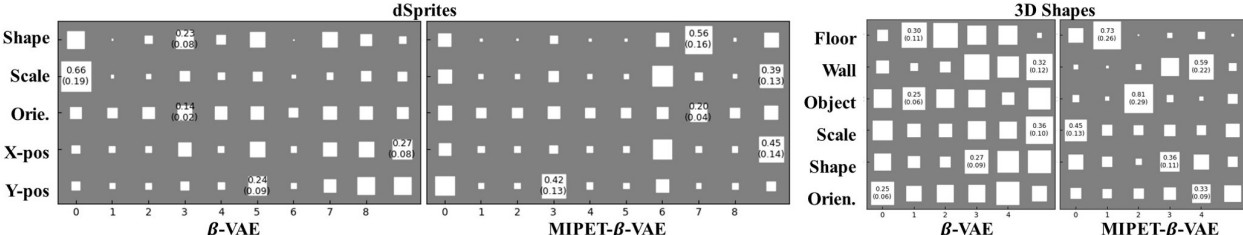

Figure 4: Each square represents a value in the DCI matrix, which describes the relationship between the $i^{th}$ latent dimension and each factor. The size of each square is relative to the values within each row. The ideal case resembles a sparse matrix. The y-axis represents the factors of each dataset, while the x-axis corresponds to the latent vector dimensions. The number shown in each row of the matrix indicates the maximum value and standard deviation of that row. Higher maximum and standard deviation values suggest greater sparsity, indicating closer alignment with the ideal case. More details in Appendix D.3.

Table 4: Impact of the number of MIPE-transformation function on the $\beta$-TCVAE and $\beta$-VAE with dSprites, 3D Shapes, and 3D Cars datasets in terms of the four metrics. The blue and red box plots represent each model's single and multiple IPE-transformation cases, respectively. (A-$n$: MIPET-$\beta$-TCVAE (4), B-$n$: MIPET-$\beta$-TCVAE (6), C-$n$: MIPET-$\beta$-VAE, $n$: the number of MIPE-transformation).

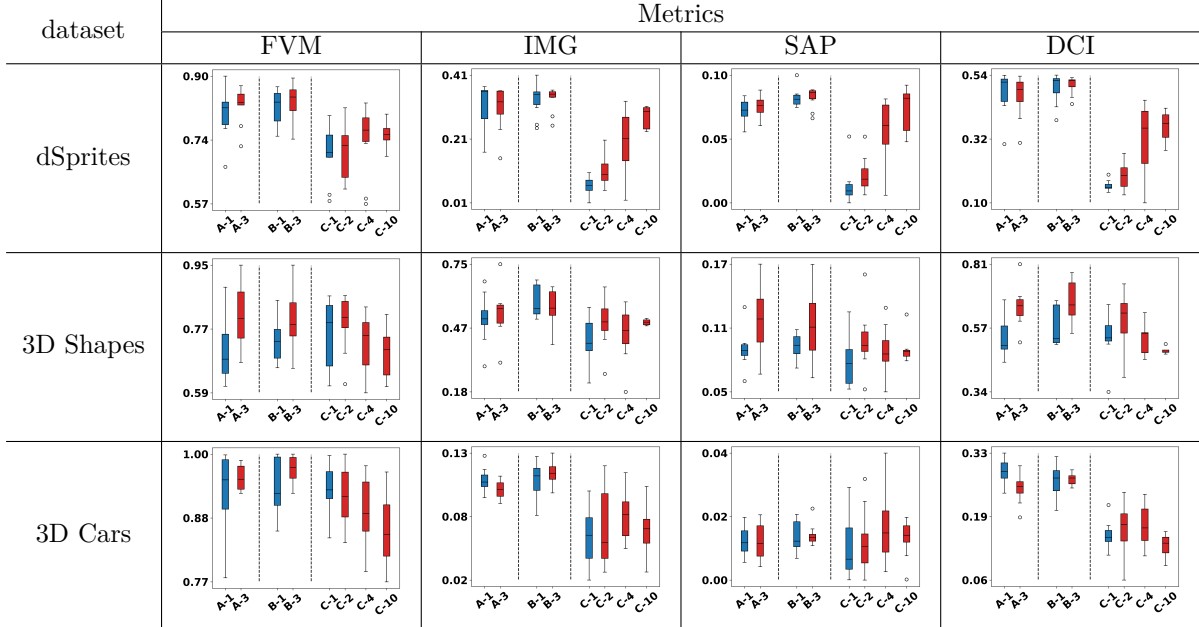

# 5 Results and Discussion

## 5.1 Quantitative Analysis

### 5.1.1 Disentanglement Metrics

We set the number of IPE-transformation functions to be equal to balancing hyper-parameter $\beta$ on $\beta$-VAE because of Eq. 11. The number of IPE-transform functions of $\beta$-TCVAE is 3. However, in the case of CLG-VAE, we set it to 1 because its approach is based on the group theory, not directly controlling a KL divergence term such as $\beta$-VAE. We average each model performance value with 40, 20, 60, 10, and 30 cases in VAEs, $\beta$-TCVAEs, Control-VAE, Factor-VAE and CLG-VAEs, respectively.

Table 5: Impact of the mask (mean±std.) and its ratio $\lambda$ in Eq. 8 on 3D Cars. ($\infty$: no masking case, gray box: the best setting over all metrics, bold text: the best in each metric.) Each model runs with ten random seeds.

| ratio | β-VAE (1) | | | | CLG-VAE (0.5) | | | |
|---|---|---|---|---|---|---|---|---|
| $\lambda$ | FVM ↑ | MIG ↑ | SAP ↑ | DCI ↑ | FVM ↑ | MIG ↑ | SAP ↑ | DIC ↑ |
| 0.0 | 90.46(±6.50) | 4.84(±2.32) | 1.29(±0.81) | 16.76(±4.68) | **90.06**(±4.44) | **9.28**(±2.09) | 1.82(±0.82) | **19.12**(±3.41) |
| 0.5 | 91.35(±5.52) | 5.37(±2.74) | 1.17(±0.67) | 16.65(±3.76) | 88.69(±4.78) | 6.90(±1.96) | 1.85(±0.67) | 17.52(±3.16) |
| 1.0 | **91.78**(±6.20) | 4.99(±2.27) | 1.36(±0.81) | 16.50(±2.53) | 83.60(±11.48) | 8.12(±3.66) | **2.37**(±1.50) | 17.07(±3.89) |
| 1.5 | 90.04(±5.88) | **7.22**(±2.87) | **1.36**(±0.48) | **18.23**(±2.84) | 84.76(±6.86) | 7.70(±2.11) | 2.05(±0.73) | 17.06(±2.77) |
| 2.0 | 87.79(±8.88) | 4.75(±2.49) | 1.01(±0.99) | 16.64(±3.75) | 85.78(±4.18) | 7.83(±1.79) | 1.91(±0.96) | 17.26(±2.07) |
| $\infty$ | 89.43(±11.72) | 3.74(±2.32) | 0.77(±0.39) | 15.45(±4.59) | 82.96(±11.84) | 8.07(±2.52) | 2.32(±1.02) | 17.46(±4.07) |

Table 6: Ablation study for the equivariant property (w/o E), and EF-conversion (w/o EF). Each metric is averaged over 40 and 20 settings of $\beta$-VAE and $\beta$-TCVAE, respectively.

| 3D Shapes | β-VAE | | | β-TCVAE | | |
|---|---|---|---|---|---|---|
| | MIPET | MIPET (w/o E) | MIPET (w/o EF) | MIPET | MIPET (w/o E) | MIPET (w/o EF) |
| FVM | **75.19**(±8.16) | 74.91(±10.46) | 22.27(±1.29) | **80.59**(±8.57) | 77.90(±8.66) | 66.38(±7.57) |
| MIG | 47.37(±10.13) | **47.45**(±8.98) | 0.28(±0.09) | **54.49**(±9.44) | 51.37(±11.54) | 36.08(±17.42) |
| SAP | 9.20(±2.44) | **9.43**(±2.59) | 0.26(±0.07) | **11.58**(±3.32) | 10.23(±3.13) | 7.13(±3.09) |
| DCI | **54.95**(±8.99) | 54.23(±9.05) | 0.10(±0.02) | **66.22**(±7.32) | 61.18(±8.87) | 56.85(±11.72) |

As shown in Table 2, MIPET-VAEs disentanglement performance is broadly improved with four metrics on each dataset. In particular, most FVM results significantly affect the model performance and stability on all datasets. Therefore, our proposed method obtains a specific dimension that corresponds to a specific single factor. These results imply that applied to MIPE-transformation functions on VAEs elaborate disentangled representation learning.

**$p$-Value** We additionally estimate the $p$-value of each metrics over models in Table 3. Previous work shows the average case of each models (Yang et al., 2022). We divide each case into four categories: 1) Positive & Significant, 2) Positive & Insignificant, 3) Negative & Insignificant, and 4) Negative & Significant, where positive is when the mean value is higher than baseline and significant is statistically significant. We estimate the probability of each category: 1) 50%, 2) 36.11%, and 3) 13.89%. As shown in Table 3 and the results, half of the cases are statistically significant, and 86.11% of cases are improved model performance. Even though our method shows a lower value than the baseline, it is not significantly decreased (13.89%). In addition, averaged results show that our method impacts to model itself without hyper-parameter tuning. $\beta$-TCVAEs is partially using our method (paragraph Models in Section 4), so it does not show the whole effect of MIPET, but it improves model performance in many cases.

**Relation Between Factors and Latent Vector Dimensions.** As shown in Fig. 4, the MIPET model shows the larger maximum value and standard deviation of each row except for the scale factor in the dSprites dataset. Our model shows a close to sparse matrix compared to the $\beta$-VAE result in 3D Shapes, also the maximum value and standard deviation of each row are larger than $\beta$-VAE. It indicates that the result of our model is close to the sparse matrix (ideal case). More details are in Appendix D.3.

### 5.1.2 Sensitivity to the Number of IPE-transformation and EF-conversion

We analyze the impact of the MIPE-transformation function. As presented in Table 4, MIPE-transformation outperforms IPE-transformation in disentanglement learning across all datasets. Indeed, the results of MIPET-$\beta$-VAEs more generally and clearly demonstrate the impact of the MIPE-transfomation function. Our derivation in Section C.1 clearly explains the impact of MIPE-transformation. This result shows the impact of the multiple uses of IPE-transformation and EF-conversion.

### 5.1.3 Impact of Implicit Semantic Mask

We set masking hyper-parameter $\lambda$ from $\{0.0, 0.5, \cdots, 2.0, \infty\}$, and each model has different $\lambda$ for best case. In Table 5, VAE and CLG-VAE with masked log-normalizer show better and well-balanced results than the models without masking, which implies improvement of disentanglement.

### 5.1.4 Ablation Study

We conduct an ablation study to evaluate the separate impact of equivariant property and the EF-conversion. We have already presented the impact of the multiple uses of IPE-transform and EF-conversion in the previous paragraph. We evaluate the impact of the other properties by setting MIPE-transformation 1) without equivariant (w/o E), which is implemented as an asymmetric matrix, and 2) without EF-conversion (w/o EF). To exclude group theory interference with other methods, we select $\beta$-VAE and $\beta$-TCVAE. As the results are shown in Table 6, most of the results show that MIPET-VAEs performance is better than other cases. In particular, MIPET (w/o EF) results are lower than MIPET (w/o E) results and are clearly shown in all cases.

## 5.2 Qualitative Analysis

### 5.2.1 Does IPE-transformation map the Gaussian Distribution to Diverse Exponential Family?

(15) As shown in Fig. 5, the posterior of VAE is a Guassian distribution. Also, the posterior of MIPET-VAE (w/o semantic mask) is close to a Gaussian distribution rather than a beta distribution. Even though IPE-transformation is a linear transformation, our model approximates the beta distribution because of the semantic mask. As we define the decoder as a single linear layer, the posterior of MIPET-VAE is close to a beta distribution. Because the linear transformation of a beta distribution is also a beta distribution. Also, MIPET-VAE preserves the dataset structure better than VAE. More details are in the Appendix D.6.

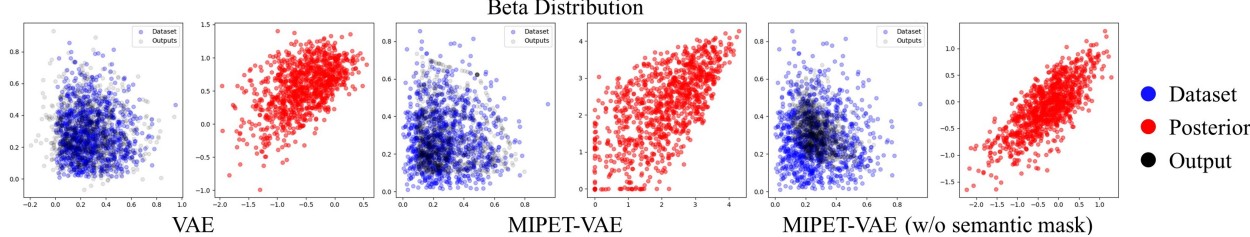

Figure 5: A toy test compares VAE, MIPET-VAE, and MIPET-VAE without semantic mask to investigate whether the model approximates the beta distribution. We construct the VAE with a 4-layer Multi-Layer Perceptron (MLP) as the encoder and a single linear layer as the decoder. Blue plots are randomly sampled from a two-dimensional beta distribution, red plots are the posterior, and black plots are the output results.

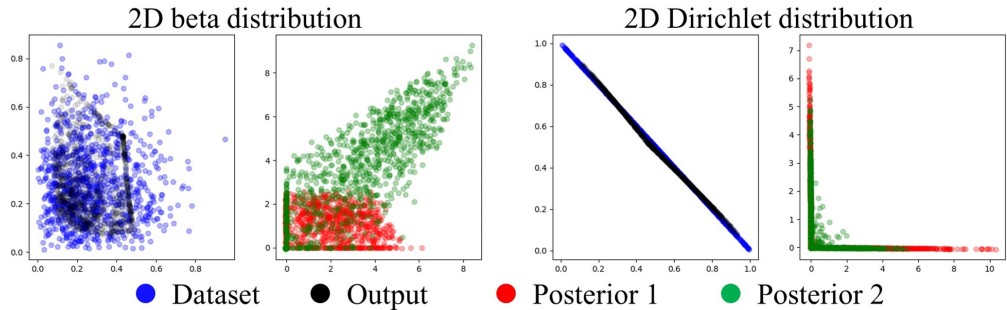

Figure 6: A toy test with two-dimensional exponential family distribution (beta and Dirichlet distribution). We set the number of IPE-transformation as 2. Green and red plots refer to latent vectors (posterior) from each IPE-transformation.

### 5.2.2 Does Each IPE-transformation map into a Different Distribution?

(16) We demonstrate how each IPE-transformation represents the exponential family. As shown in Fig. 6, each IPE-transformation represents a different distribution because the green and red plots are distributed in different posteriors, even though all IPE-transformations share the same decoder weights.

### 5.2.3 Image Generation

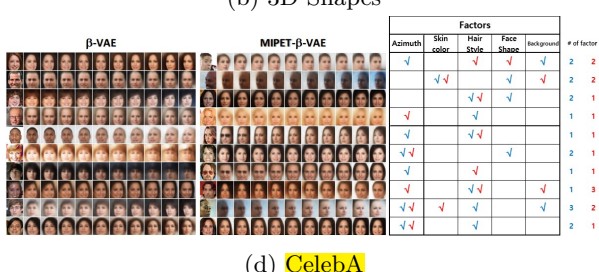

(a) dSprites

(b) 3D Shapes

(c) 3D Car

(d) CelebA

Figure 7: Qualitative results on dSprites. The left-side grids are input images and their variants by changing activations of each dimension of latent vectors. The first row shows input images. The right-side table shows matching pre-defined factors of the dataset (red: MIPET, blue: no MIPET).

We randomly sample an image for each dimension of the latent vector space and creates 10 variants of its generated latent vector by selecting values from {-2, 2} with 10 intervals for the dimension, then generate their corresponding output images. For the generation, we select $\beta$-TCVAE (6), which shows the best FVM scores in dSprites dataset. Thereafter, we evaluate the semantic roles of each dimension before and after applying MIPE-transformation function.

In Figure 7a, $\beta$-TCVAE struggles with y-position and rotation, as shown on the $6^{th}$ row, and with scale and shape represented on the $7^{th}$ row. On the contrary, MIPET-$\beta$-TCVAE separates y-position and rotation factor ($10^{th}$, and $7^{th}$ rows), also the activated dimensions of MIPET-$\beta$-TCVAE are not overlapped with each factor. Applied our method on $\beta$-TCVAE shows better disentangled representation on dSprites dataset. These results also show that our proposed method improves disentangled representation learning.

As shown in the Figure 7b, CLG-VAE struggles with shape and wall color factors in $4^{th}$ dimension, and shape and object color factors in $7^{th}$ dimension. In particular, it struggles with tree factors in $9^{th}$ dimension. On the other hand, MIPET-CLG-VAE separates shape, wall, and object color factors.

The qualitative analysis with 3D Cars dataset, as show in Figure 7c, the left side is the $\beta$-TCVAE result, and it struggles with body, and azimuth factors shown in the $7^{th}$ row. However, MIPET-$\beta$-TCVAE separates azimuth ($6^{th}$ row) and body ($1^{st}$ row). In particular, MIPET-$\beta$-TCVAE learns *color* factor ($3^{rd}$ row) which does not exist on $\beta$-TCVAE.

(17) Furthermore, our model shows slight improvement on the CelebA (Liu et al., 2015) dataset, as shown in Fig. 7d. The overlapping dimensions between the baseline and our model are 6 and 4, respectively.

## 6 Conclusion

In this paper, we address the problem of injecting inductive bias for learning unsupervised disentangled representations. To build the bias in VAE frameworks, we propose MIPE-transformation composed of 1) IPE-transformation for the benefits of invertibility and partial-equivariance for disentanglement, 2) a training loss and module to adapt unrestricted prior and posterior to an approximated exponential family, and 3) integration of multiple units of IPE-transformation function and EF-conversion for more expressive bias. The method is easily equipped on state-of-the-art VAEs for disentanglement learning and shows significant

improvement on dSprites, 3D Shapes, and 3D Cars datasets. We expect that our method can be applied to more VAEs, and extended to downstream applications. (18) Although our model does not show sufficiently improved results on complex datasets such as CelebA or beyond, we anticipate that utilizing architectures like CLIP (Radford et al., 2021), which directly train symmetry (transformation) information through language and vision datasets, would enable more efficient injection of inductive bias compared to unsupervised learning approaches. Our work is limited to holding partial equivariance of I2L transformation, so more direct methods to induce it can be integrated in the future.

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

# A    Appendix

# B    Preliminaries

## B.1    Group Theory

**Binary operation:** Binary operation on a set $S$ is a function that $* : S \times S \to S$, where $\times$ is a cartesian product.

**Group:** A group is a set $G$ together with binary operation $*$, that combines any two elements $g_a$ and $g_b$ in $G$, such that the following properties:

- closure: $g_a, g_b \in G \Rightarrow g_a * g_b \in G$.

- Associativity: $\forall g_a, g_b, g_c \in G$, s.t. $(g_a * g_b) * g_c = g_a * (g_b * g_c)$.

- Identity element: There exists an element $e \in G$, s.t. $\forall g \in G, e * g = g * e = g$.

- Inverse element: $\forall g \in G, \exists g^{-1} \in G$: $g * g^{-1} = g^{-1} * g = e$.

**Group action:** Let $(G, *)$ be a group and set $X$, binary operation $\cdot : G \times X \to X$, such that following properties:

- Identity: $e \cdot x = x$, where $e \in G, x \in X$.

- Compatibility: $\forall g_a, g_b \in G, x \in X, (g_a * g_b) \cdot x = g_a \cdot (g_b \cdot x)$.

**Equivariant map:** Let $G$ be a group and $X_1, X_2$ be two sets with corresponding group action of $G$ in each sets: $T_g^{X_1}, T_g^{X_2}$, where $g \in G$. Then a function $f : X_1 \to X_2$ is equivariant if $f(T_g^{X_1} \cdot X_1) = T_g^{X_2} \cdot f(X_1)$.

**Partial Equivariance:** Romero & Lohit (2022): Let subset of $G$ be $\Upsilon \subset G$, then $f$ is a partially equivariant map to $G$:

$$f(T_\upsilon^{X_1} \cdot X_1) = T_\upsilon^{X_2} \cdot f(X_1), \text{ where } \forall \upsilon \in \Upsilon. \tag{12}$$

**Homomorphsim:** Let $(G, \cdot), (H, \circ)$ be two groups. If mapping function $h : G \to H$, s.t. $h(g_i \cdot g_j) = f(g_i) \circ f(g_j)$, then $f$ is called homomorphism.

## B.2    Exponential Family

Power density function of the exponential family (PDF) generalized formulation:

$$\begin{aligned} f_{\boldsymbol{x}}(\boldsymbol{x}|\boldsymbol{\theta}) &= h(\boldsymbol{x})\exp(\boldsymbol{\theta}^\intercal T(\boldsymbol{x}) - A(\boldsymbol{\theta})) \\ &= \exp(\boldsymbol{\theta}^\intercal T(\boldsymbol{x}) - A(\boldsymbol{\theta}) + B(\boldsymbol{x})), \end{aligned} \tag{13}$$

where *sufficient statistics* $T(\cdot)$, *log-normalizer* $A(\cdot)$, and *carrier or base measure* $B(\cdot)$ are known functions, samples $\boldsymbol{x}$ from distribution, and *natural parameter* $\boldsymbol{\theta}$.

## C Proof

**Proposition C.1.** *If $q_\phi$ is equivariant over defined on group of symmetries $G_I^J$ and $G_L^J$, then $\psi_{G_S}(q_\phi(\cdot))$ is equivariant to symmetries in $G_I$ corresponding to $G_S \cap G_L$ and $G_T$ corresponding to $G_S \cap G_L$ by the equivariance of $q_\phi$.*

*Proof.* The function $\psi_{G_S}(\cdot)$ is an equivariant function over group elements in $G_S \cap G_L$ by Proposition 3.1. Then, the composite function, $\psi_{G_S}(\cdot)$ and $q_\phi$, is an equivariant function of $G_I$ corresponding to $G_S \cap G_L$ and $G_T$ corresponding to $G_S \cap G_L$. Let $g_L^J$ be a group element in $G_S \cap G_L$, and $g_I^J$ is a group element in $G_I$ corresponding to $G_S \cap G_L$, and $g_T^J$ is a group element where corresponding to $G_S \cap G_L$ on the latent vector space transformed from the original latent vector space. Then, group element $g_T^J$ is equal to $g_L^J$:

$$\hat{z}_1 = \psi_{G_S}(z_1), \text{and} \tag{14}$$

$$\hat{z}_2 = \psi_{G_S}(z_2) = \psi_{G_S}(g_L^a z_1) = g_L^a \psi_{G_S}(z_1) \ (\because \text{Prop. 3.1}), \tag{15}$$

$$\text{then } g_L^a \psi_{G_S}(z_1) = g_T^a \psi_{G_S}(z_1) \ (\because \hat{z}_2 = g_T^a \hat{z}_1)$$
$$\Rightarrow (g_L^a - g_T^a)\psi_{G_S}(z_1) = \mathbf{0}, \tag{16}$$

where $\mathbf{0}$ is a zero vector. Eq. 16 is defined when $\forall z \in \mathcal{Z}$ by the equivariance definition. In other words, Eq. 16 is satisfied only if the kernel (linear algebra) of $g_L^a - g_T^a$, notated as $ker(g_L^a - g_T^a)$, includes the basis of $\mathbb{R}^n$ vector space. If the standard basis of $\mathbb{R}^n$ vector space is in $ker(g_L^a - g_T^a)$, then $(g_L^a - g_T^a) = \mathbf{0}_{n,n}$, where $\mathbf{0}_{n,n}$ is an n by n zero matrix. Other bases of $\mathbb{R}^n$ vector space are expressed by the standard basis. Therefore $g_L^a - g_T^a = \mathbf{0}_{n,n}$.

Then, $\psi_{G_S}(g_L^a z_1) = g_L^a \psi_{G_S}(z_1) = g_T^a \psi_{G_S}(z_1)$. The encoder is an equivariant function over input space $\mathcal{X}$ as $q_\phi(g_I^a x_1) = g_L^a q_\phi(x_1)$. Mixing two equivarience property, we can derive another equivariance relation $g_T^a \psi_{G_S}(q_\phi(x_1)) = \psi_{G_S}(q_\phi(g_I^a x_1))$ This result implies that the equivariance between input space and a latent space is preserved for $G_S \cap G_L$ if the latent vector $z$ is transformed by $\psi_{G_S}$.∎

We show that $\psi_{G_S}$ preserves equivariance between $G_L^J$ and $G_I^J$. If there exists equivariant function between input and latent vector space, there should be a group $G_L$ for a latent space and its corresponding group $G_I$ in an input space by definition of equivariance ($q_\phi(g_I x) = g_L q_\phi(x)$).

In other words, $\psi_{G_S}(\cdot)$ guarantees to preserve the equivariance of I2L-transformation to certain symmetries in $G_S \cap G_L$ after IPE-transformation as shown in Figure 2.

Let $P(B)$ be the probability of $\psi(\cdot) \in B$ for a subset $B \subset M_n(\mathbb{R})$ after VAE training, and $Pr(\psi_B \in B')$ be the conditional probability of $\psi(\cdot) \in B'$ given $\psi(\cdot) \in B$.

Then,

**Proposition C.2.** $Pr(\psi_{E_S}(\cdot) \in G_S) > Pr(\psi_{E_M}(\cdot) \in G_S) > Pr(\psi_M(\cdot) \in G_S)$.

*Proof.* All $\mathbf{e}^S \in E_S$ are in $E_M$ since $Sym_n(\mathbb{R}) \subset M_n(\mathbb{R})$. However, $E_M \not\subset E_S$ because $\mathbf{e}^S$ is always symmetric, but $\mathbf{e}^M$ can be an asymmetric matrix. All elements of $E_S$ are symmetric because of the matrix exponential property that $\mathbf{e}^{M^\intercal} = (\mathbf{e}^M)^\intercal$. If $M$ is a symmetric matrix then $\mathbf{e}^{M^\intercal} = \mathbf{e}^M = (\mathbf{e}^M)^\intercal$. Therefore, if $M$ is symmetric then the exponential of $M$ is also symmetric. We show a counter example to $E_M \subset E_S$. When $M = \begin{bmatrix} 1 & 1 \\ 0 & 1 \end{bmatrix}$,

$$\mathbf{e}^M = \sum_{k=0}^{\infty} \frac{1}{k!} M^k$$

$$= I + \begin{bmatrix} 1 & 1 \\ 0 & 1 \end{bmatrix} + \frac{1}{2!}\begin{bmatrix} 1 & 1 \\ 0 & 1 \end{bmatrix}^2 + \cdots + \frac{1}{(n-1)!}\begin{bmatrix} 1 & 1 \\ 0 & 1 \end{bmatrix}^{(n-1)} + \cdots$$

$$= I + \begin{bmatrix} \sum_{n=0}^{\infty} \frac{1}{n!} & 1 + \sum_{n=0}^{\infty} \frac{1}{(n-1)!} \\ 0 & \sum_{n=0}^{\infty} \frac{1}{n!} \end{bmatrix}$$

$$= \begin{bmatrix} 1 + e & 1 + e \\ 0 & 1 + e \end{bmatrix}. \tag{17}$$

The matrix $\mathbf{e}^{\boldsymbol{M}}$ is asymmetric and not in $E_S$. Therefore $E_M \not\subset E_S$. Therefore, the probability $Pr(\psi_{E_S}(\cdot) \in G_S) = \frac{P(G_S)}{P(E_S)}$ is greater than $Pr(\psi_{E_M}(\cdot) \in G_S) = \frac{P(G_S)}{P(E_M)}$. In the same way, $Pr(\psi_{E_M}(\cdot) \in G_S) > Pr(\psi_M(\cdot) \in G_S) = \frac{P(G_S)}{P(M_n(\mathbb{R}))}$ because $E_M \subset M_n(\mathbb{R})$ and non-invertible functions are only in $M_n(\mathbb{R})$. $\blacksquare$

### C.0.1   KL Divergence for Evidence of Lower Bound

The KL divergence of Gaussian distribution Kingma & Welling (2013) is computed using mean and variance, which are the parameters of a Gaussian distribution. To introduce a loss as the KL divergence of Gaussian distribution, we compute KL divergence of the exponential family in Eq. 2 using the learnable parameter $T(\cdot)$ and $A(\cdot)$ with given natural parameter $\boldsymbol{\theta}_{\hat{\boldsymbol{z}}}$ and $\boldsymbol{\theta}_{\hat{\boldsymbol{\epsilon}}}$, expressed as:

$$
\begin{aligned}
\mathcal{L}_{kl} &:= D_{\mathrm{KL}}(f_{\boldsymbol{x}}(\boldsymbol{x}|\boldsymbol{\theta}_{\hat{\boldsymbol{z}}_m})||f_{\boldsymbol{x}}(\boldsymbol{x}|\boldsymbol{\theta}_{\hat{\boldsymbol{\epsilon}}_m})) \\
&= A(\boldsymbol{\theta}_{\boldsymbol{\epsilon}}) - A(\boldsymbol{\theta}_{\hat{\boldsymbol{z}}}) + \boldsymbol{\theta}_{\hat{\boldsymbol{z}}}^{\mathsf{T}}\nabla_{\boldsymbol{\theta}_{\hat{\boldsymbol{z}}}}A(\boldsymbol{\theta}_{\hat{\boldsymbol{z}}}) - \boldsymbol{\theta}_{\hat{\boldsymbol{\epsilon}}}^{\mathsf{T}}\nabla_{\boldsymbol{\theta}_{\boldsymbol{\epsilon}}}A(\boldsymbol{\theta}_{\boldsymbol{\epsilon}}).
\end{aligned}
\tag{18}
$$

Because $D_{\mathrm{KL}}(f_{\boldsymbol{x}}(\boldsymbol{x}|\boldsymbol{\theta}_{\hat{\boldsymbol{z}}})||f_{\boldsymbol{x}}(\boldsymbol{x}|\boldsymbol{\theta}_{\hat{\boldsymbol{\epsilon}}}))$ is followed as:

$$
\begin{aligned}
D_{\mathrm{KL}}(f_{\boldsymbol{x}}(\boldsymbol{x}|\boldsymbol{\theta}_{\hat{\boldsymbol{z}}})||f_{\boldsymbol{x}}(\boldsymbol{x}|\boldsymbol{\theta}_{\hat{\boldsymbol{\epsilon}}})) &= \int_{-\infty}^{\infty} f_{\boldsymbol{x}}(\boldsymbol{x}|\boldsymbol{\theta}_{\hat{\boldsymbol{z}}})\log f_{\boldsymbol{x}}(\boldsymbol{x}|\boldsymbol{\theta}_{\hat{\boldsymbol{z}}})\mathrm{d}\boldsymbol{x} \\
&\quad - \int_{-\infty}^{\infty} f_{\boldsymbol{x}}(\boldsymbol{x}|\boldsymbol{\theta}_{\hat{\boldsymbol{z}}})\log f_{\boldsymbol{x}}(\boldsymbol{x}|\boldsymbol{\theta}_{\hat{\boldsymbol{\epsilon}}})\mathrm{d}\boldsymbol{x}.
\end{aligned}
\tag{19}
$$

We designed sufficient statistics as matrix multiplication (multi-layer perceptron). Then,

$$
\begin{aligned}
\int_{-\infty}^{\infty} f_{\boldsymbol{x}}(\boldsymbol{x}|\boldsymbol{\theta}_{\hat{\boldsymbol{z}}})\log f_{\boldsymbol{x}}(\boldsymbol{x}|\boldsymbol{\theta}_{\hat{\boldsymbol{z}}})\mathrm{d}\boldsymbol{x} &= \int_{-\infty}^{\infty} f_{\boldsymbol{x}}(\boldsymbol{x}|\boldsymbol{\theta}_{\hat{\boldsymbol{z}}})\cdot \\
&\qquad [\boldsymbol{\theta}_{\hat{\boldsymbol{z}}}^{\mathsf{T}}\mathbf{T}(\boldsymbol{x}) - A(\boldsymbol{\theta}_{\hat{\boldsymbol{z}}}) + B(\boldsymbol{x})]\mathrm{d}\boldsymbol{x} \\
&= -A(\boldsymbol{\theta}_{\hat{\boldsymbol{z}}})\int_{-\infty}^{\infty} f_{\boldsymbol{x}}(\boldsymbol{x}|\boldsymbol{\theta}_{\hat{\boldsymbol{z}}})\mathrm{d}\boldsymbol{x} \\
&\quad + \int_{-\infty}^{\infty} f_{\boldsymbol{x}}(\boldsymbol{x}|\boldsymbol{\theta}_{\hat{\boldsymbol{z}}})[\boldsymbol{\theta}_{\hat{\boldsymbol{z}}}^{\mathsf{T}}\mathbf{T}(\boldsymbol{x}) + B(\boldsymbol{x})]\mathrm{d}\boldsymbol{x} \\
&= -A(\boldsymbol{\theta}_{\hat{\boldsymbol{z}}}) + \boldsymbol{\theta}_{\hat{\boldsymbol{z}}}^{\mathsf{T}}\int_{-\infty}^{\infty} T(\boldsymbol{x})f_{\boldsymbol{x}}(\boldsymbol{x}|\boldsymbol{\theta}_{\hat{\boldsymbol{z}}})\mathrm{d}\boldsymbol{x} \\
&\quad + \int_{-\infty}^{\infty} B(\boldsymbol{x})f_{\boldsymbol{x}}(\boldsymbol{x}|\boldsymbol{\theta}_{\hat{\boldsymbol{z}}})\mathrm{d}\boldsymbol{x},
\end{aligned}
\tag{20}
$$

and

$$
\begin{aligned}
\int_{-\infty}^{\infty} f_{\boldsymbol{x}}(\boldsymbol{x}|\boldsymbol{\theta}_{\hat{\boldsymbol{z}}})\log f_{\boldsymbol{x}}(\boldsymbol{x}|\boldsymbol{\theta}_{\hat{\boldsymbol{\epsilon}}})\mathrm{d}\boldsymbol{x} &= -A(\boldsymbol{\theta}_{\hat{\boldsymbol{\epsilon}}}) \\
&\quad + \boldsymbol{\theta}_{\hat{\boldsymbol{\epsilon}}}^{\mathsf{T}}\int_{-\infty}^{\infty} T(\boldsymbol{x})f_{\boldsymbol{x}}(\boldsymbol{x}|\boldsymbol{\theta}_{\hat{\boldsymbol{\epsilon}}})\mathrm{d}\boldsymbol{x} \\
&\quad + \int_{-\infty}^{\infty} B(\boldsymbol{x})f_{\boldsymbol{x}}(\boldsymbol{x}|\boldsymbol{\theta}_{\hat{\boldsymbol{\epsilon}}})\mathrm{d}\boldsymbol{x}.
\end{aligned}
\tag{21}
$$

$$
\begin{aligned}
\therefore D_{\mathrm{KL}}(f_{\boldsymbol{x}}(\boldsymbol{x}|\boldsymbol{\theta}_{\hat{\boldsymbol{z}}})||f_{\boldsymbol{x}}(\boldsymbol{x}|\boldsymbol{\theta}_{\hat{\boldsymbol{\epsilon}}})) &= A(\boldsymbol{\theta}_{\hat{\boldsymbol{\epsilon}}}) - A(\boldsymbol{\theta}_{\hat{\boldsymbol{z}}}) \\
&\quad + \boldsymbol{\theta}_{\hat{\boldsymbol{z}}}\int_{-\infty}^{\infty} T(\boldsymbol{x})f_{\boldsymbol{x}}(\boldsymbol{x}|\boldsymbol{\theta}_{\hat{\boldsymbol{z}}})\mathrm{d}\boldsymbol{x} \\
&\quad - \boldsymbol{\theta}_{\hat{\boldsymbol{\epsilon}}}\int_{-\infty}^{\infty} T(\boldsymbol{x})f_{\boldsymbol{x}}(\boldsymbol{x}|\boldsymbol{\theta}_{\hat{\boldsymbol{\epsilon}}})\mathrm{d}\boldsymbol{x}.
\end{aligned}
\tag{22}
$$

The mean of the sufficient statistic is followed as:

$$\int_{-\infty}^{\infty} T(\boldsymbol{x}) f_{\boldsymbol{x}}(\boldsymbol{x}|\boldsymbol{\theta}) \mathrm{d}\boldsymbol{x} = \frac{\partial A^*(\boldsymbol{\theta})}{\partial \boldsymbol{\theta}} \approx \frac{\partial A(\boldsymbol{\theta})}{\partial \boldsymbol{\theta}} \qquad \because A^*(\boldsymbol{\theta}) = \boldsymbol{\theta}^{\mathsf{T}} A^*, \tag{23}$$

where $A^*(\cdot)$ is a true log-partition function of the exponential family (ideal case of $A(\cdot)$). However, estimating $A^*$ is difficult, and there is no direct method without random samplings, such as mini-batch weighted sampling or mini-batch stratified sampling Chen et al. (2018). Then, we approximate $A^*$ to $A$, and train $A$ to be close to $A^*$. Consequently, we obtain KL divergence of the exponential family as:

$$\int_{-\infty}^{\infty} f_{\boldsymbol{x}}(\mathbf{x}|\boldsymbol{\theta}_{\hat{\boldsymbol{z}}}) \log f_{\boldsymbol{x}}(\boldsymbol{x}|\boldsymbol{\theta}_{\hat{\boldsymbol{z}}}) = -A(\boldsymbol{\theta}_{\hat{\boldsymbol{z}}}) + \boldsymbol{\theta}_{\hat{\boldsymbol{z}}}^{\mathsf{T}} \frac{\partial A(\boldsymbol{\theta}_{\hat{\boldsymbol{z}}})}{\partial \boldsymbol{\theta}_{\hat{\boldsymbol{z}}}}$$

$$+ \int_{-\infty}^{\infty} f_{\boldsymbol{x}}(\boldsymbol{x}|\boldsymbol{\theta}_{\hat{\boldsymbol{z}}}) B(\boldsymbol{x}) \mathrm{d}\boldsymbol{x}, \tag{24}$$

$$\int_{-\infty}^{\infty} f_{\boldsymbol{x}}(\boldsymbol{x}|\boldsymbol{\theta}_{\hat{\boldsymbol{z}}}) \log f_{\boldsymbol{x}}(\mathbf{x}|\boldsymbol{\theta}_{\hat{\boldsymbol{\epsilon}}}) = -Z(\boldsymbol{\theta}_{\hat{\boldsymbol{\epsilon}}}) + \boldsymbol{\theta}_{\hat{\boldsymbol{\epsilon}}}^{\mathsf{T}} \frac{\partial A(\boldsymbol{\theta}_{\hat{\boldsymbol{\epsilon}}})}{\partial \boldsymbol{\theta}_{\hat{\boldsymbol{\epsilon}}}}$$

$$+ \int_{-\infty}^{\infty} f_{\boldsymbol{x}}(\boldsymbol{x}|\boldsymbol{\theta}_{\hat{\boldsymbol{z}}}) B(\boldsymbol{x}) \mathrm{d}\boldsymbol{x}. \tag{25}$$

Therefore, the final Kullback-Leibler divergence of exponential family is followed as:

$$D_{\mathrm{KL}}(f_{\boldsymbol{x}}(\boldsymbol{x}|\boldsymbol{\theta}_{\hat{\boldsymbol{z}}})||f_{\boldsymbol{x}}(\boldsymbol{x}|\boldsymbol{\theta}_{\hat{\boldsymbol{\epsilon}}})) = A(\boldsymbol{\theta}_{\hat{\boldsymbol{\epsilon}}}) - A(\boldsymbol{\theta}_{\hat{\boldsymbol{z}}}) + \boldsymbol{\theta}_{\hat{\boldsymbol{z}}}^{\mathsf{T}} \frac{\partial A(\boldsymbol{\theta}_{\hat{\boldsymbol{z}}})}{\partial \boldsymbol{\theta}_{\hat{\boldsymbol{z}}}} - \boldsymbol{\theta}_{\hat{\boldsymbol{\epsilon}}}^{\mathsf{T}} \frac{\partial A(\boldsymbol{\theta}_{\hat{\boldsymbol{\epsilon}}})}{\partial \boldsymbol{\theta}_{\hat{\boldsymbol{\epsilon}}}}. \tag{26}$$

### C.1 Integration for Multiple IPE-Transformation and EF-Conversion

We mathematically extend IPE-transformation to MIPE-transformation, which is the equivalent process of $\beta$-VAE to enhance disentanglement. Each IPE-transformation function operates independently, then the reconstruction error for objective function is defined as:

$$\mathcal{L}_{rec} := \frac{1}{k} \sum_{i=1}^{k} \left[ \int q_i(\hat{\boldsymbol{z}}_i|\boldsymbol{x}) \log p_\theta(\boldsymbol{x}|\hat{\boldsymbol{z}}_i) \mathrm{d}\hat{\boldsymbol{z}}_i \prod_{j=1, j\neq i}^{k} \int q_j(\hat{\boldsymbol{z}}_j|\boldsymbol{x}) \mathrm{d}\hat{\boldsymbol{z}}_j \right]$$

$$= \frac{1}{k} \sum_{i=1}^{k} E_{q_{\phi,\psi_i}(\boldsymbol{z}|\boldsymbol{x})} \log p_\theta(\boldsymbol{x}|\psi_i(\boldsymbol{z})), \tag{27}$$

where $\hat{\boldsymbol{z}}_i = \psi_i(\boldsymbol{z})$. Becuase the log likelihood of $p(\boldsymbol{x})$ can be derived as follows:

$$\log p_\theta(\boldsymbol{x}) = \int \prod_i^k q_1(\hat{\boldsymbol{z}}_i|\boldsymbol{x}) \log p_\theta(\boldsymbol{x}) \mathrm{d}\hat{\boldsymbol{z}}' \tag{28}$$

$$= \int \prod_i^k q_1(\hat{\boldsymbol{z}}_i|\boldsymbol{x}) \log \frac{p_\theta(\boldsymbol{x}, \hat{\boldsymbol{z}}_1, \hat{\boldsymbol{z}}_2, \cdots, \hat{\boldsymbol{z}}_k)}{p_\theta(\hat{\boldsymbol{z}}_1, \hat{\boldsymbol{z}}_2, \cdots, \hat{\boldsymbol{z}}_k|\boldsymbol{x})} \mathrm{d}\hat{\boldsymbol{z}}' \tag{29}$$

$$= \int \prod_i^k q_1(\hat{\boldsymbol{z}}_i|\boldsymbol{x}) \cdot$$

$$\left[ \log \frac{p_\theta(\boldsymbol{x}, \hat{\boldsymbol{z}}_1, \hat{\boldsymbol{z}}_2, \cdots, \hat{\boldsymbol{z}}_k)}{q(\hat{\boldsymbol{z}}_1, \hat{\boldsymbol{z}}_2, \cdots, \hat{\boldsymbol{z}}_k|\boldsymbol{x})} - \log \frac{p_\theta(\hat{\boldsymbol{z}}_1, \hat{\boldsymbol{z}}_2, \cdots, \hat{\boldsymbol{z}}_k|\boldsymbol{x})}{q(\hat{\boldsymbol{z}}_1, \hat{\boldsymbol{z}}_2, \cdots, \hat{\boldsymbol{z}}_k|\boldsymbol{x})} \right] \mathrm{d}\hat{\boldsymbol{z}}' \tag{30}$$

$$\geqq \int \prod_i^k q_1(\hat{\boldsymbol{z}}_i|\boldsymbol{x}) \log \frac{p_\theta(\boldsymbol{x}, \hat{\boldsymbol{z}}_1, \hat{\boldsymbol{z}}_2, \cdots, \hat{\boldsymbol{z}}_k)}{q(\hat{\boldsymbol{z}}_1, \hat{\boldsymbol{z}}_2, \cdots, \hat{\boldsymbol{z}}_k|\boldsymbol{x})} \mathrm{d}\hat{\boldsymbol{z}}' \tag{31}$$

$$= \int \prod_i^k q_1(\hat{\boldsymbol{z}}_i|\boldsymbol{x}) \cdot$$

$$\left[ \log p_\theta(\boldsymbol{x}|\hat{\boldsymbol{z}}_1, \hat{\boldsymbol{z}}_2, \cdots, \hat{\boldsymbol{z}}_k) + \log \frac{p(\hat{\boldsymbol{z}}_1, \hat{\boldsymbol{z}}_2, \cdots, \hat{\boldsymbol{z}}_k)}{q(\hat{\boldsymbol{z}}_1, \hat{\boldsymbol{z}}_2, \cdots, \hat{\boldsymbol{z}}_k|\boldsymbol{x})} \right] \mathrm{d}\hat{\boldsymbol{z}}', \tag{32}$$

where $\mathrm{d}\hat{z}' = \mathrm{d}\hat{z}_1 \mathrm{d}\hat{z}_2 \cdots \mathrm{d}\hat{z}_k$. Each IPE-transformation function operates independently, then $\log p_\theta(\boldsymbol{x}|\hat{z}_1, \hat{z}_2, \cdots, \hat{z}_k) = -(k-1)\log p_\theta(\boldsymbol{x}) + \Pi_{i=1}^k p_\theta(\boldsymbol{x}|\hat{z}_i)$. Then,

$$
\begin{aligned}
p_\theta(\boldsymbol{x}|\hat{z}_1, \hat{z}_2, \ldots, \hat{z}_k) &= \frac{p_\theta(\hat{z}_1, \hat{z}_2, \ldots, \hat{z}_k|\boldsymbol{x})p_\theta(\boldsymbol{x})}{p_\theta(\hat{z}_1, \hat{z}_2, \ldots, \hat{z}_k)} \\
&= \frac{p_\theta(\boldsymbol{x}) \prod_{i=1}^k p_\theta(\hat{z}_i|\boldsymbol{x})}{\prod_{i=1}^k p_\theta(\hat{z}_i)} (\because (\hat{z}_i \perp\!\!\!\perp \hat{z}_j s|\boldsymbol{x})) \\
&= \prod_{i=1}^k \frac{p_\theta(\hat{z}_i|\boldsymbol{x})p_\theta(\boldsymbol{x}^{\frac{1}{k}})}{p_\theta(\hat{z}_i)} \\
&= p_\theta(\boldsymbol{x})^{-(k-1)} \prod_{i=1}^k \frac{p_\theta(\hat{z}_i|\boldsymbol{x})p_\theta(\boldsymbol{x})}{p_\theta(\hat{z}_i)} \\
&= p_\theta(\boldsymbol{x})^{-(k-1)} \prod_{i=1}^k p_\theta(\boldsymbol{x}|\hat{z}_i),
\end{aligned}
\tag{33}
$$

where $\hat{z}_j s = \cap_{j=1, j \neq i}^k \hat{z}_j$. Therefore,

$$
\begin{aligned}
&\int \prod_{i=1}^k q_i(\hat{z}_i|\boldsymbol{x}) \log p_\theta(\boldsymbol{x}|\hat{z}_1, \hat{z}_2, \cdots, \hat{z}_k) \mathrm{d}\hat{z}' \\
&= \int \prod_{i=1}^k q_i(\hat{z}_i|\boldsymbol{x}) \Big[ -(k-1)\log p_\theta(\boldsymbol{x}) + \prod_{i=1}^k p_\theta(\boldsymbol{x}|\hat{z}_i) \Big] \mathrm{d}\hat{z}' \\
&= -(k-1)\log p_\theta(\boldsymbol{x}) + \int \prod_{i=1}^k q_i(\hat{z}_i|\boldsymbol{x}) \prod_{i=j}^k p_\theta(\boldsymbol{x}|\hat{z}_j) \mathrm{d}\hat{z}'.
\end{aligned}
\tag{34}
$$

Then,

$$
\begin{aligned}
\log p_\theta(\boldsymbol{x}) &\geq \frac{1}{k} \sum_{i=1}^k \left[ \int q_i(\hat{z}_i|\boldsymbol{x}) \log p_\theta(\boldsymbol{x}|\hat{z}_i) \mathrm{d}\hat{z}_i \prod_{j=1, j \neq i}^k \int q_j(\hat{z}_j|\boldsymbol{x}) \mathrm{d}\hat{z}_j \right] \\
&\quad - \int \prod_i^k q_1(\hat{z}_i|\boldsymbol{x}) \log \frac{\prod_{i=1}^k q_i(\hat{z}_i|\boldsymbol{x})}{\prod_{i=1}^k p(\hat{z}_i)} \mathrm{d}\hat{z}' \\
&= \frac{1}{k} \sum_{i=1}^k \mathbb{E}_{q(\hat{z}_i|\boldsymbol{x})} \log p_\theta(\boldsymbol{x}|\hat{z}_i) \\
&\quad - \sum_{i=1}^k \left[ D_{\mathrm{KL}}(q_\phi(\hat{z}_i|\boldsymbol{x})||p(\hat{z}_i)) \prod_{j=1, j \neq i}^k \int q_j(\hat{z}_j|\boldsymbol{x}) \mathrm{d}\hat{z}_j \right] \\
&= \frac{1}{k} \sum_{i=1}^k \mathbb{E}_{q_\phi(\hat{z}_i|\boldsymbol{x})} \log p_\theta(\boldsymbol{x}|\hat{z}_i) - \sum_{i=1}^k D_{\mathrm{KL}}(q_\phi(\hat{z}_i|\boldsymbol{x})||p(\boldsymbol{z}_i)) \\
&= \frac{1}{k} \Big[ \sum_{i=1}^k \mathbb{E}_{q_\phi(\hat{z}_i|\boldsymbol{x})} \log p_\theta(\boldsymbol{x}|\hat{z}_i) - k D_{\mathrm{KL}}(q_\phi(\hat{z}_i|\boldsymbol{x})||p(\hat{z}_i)) \Big].
\end{aligned}
\tag{35}
$$

Therefore, we define ELBO as:

$$
\begin{aligned}
\mathcal{L}'(\phi, \theta, \psi_{i \in [1,k]}; \boldsymbol{x}) = {} & \frac{1}{k} \sum_{i=1}^k \mathbb{E}_{q_{\phi,\psi_i}(\boldsymbol{z}_i|\boldsymbol{x})} \log p_\theta(\boldsymbol{x}|\psi_i(\boldsymbol{z})) - \\
& \sum_{i=1}^k D_{\mathrm{KL}}(q_{\phi,\psi_i}(\boldsymbol{z}|\boldsymbol{x})||p_{\psi_i}(\boldsymbol{z})).
\end{aligned}
\tag{36}
$$

However, following Eq. 36, k samples are generated, and each sample is disentangled for different factors. We implement the output as the average of the sum of the k samples to obtain a single sample with a superposition effect from k samples. Moreover, the KL divergence term in Eq. 36 represents that increasing number of MIPE-transformation is equal to increasing $\beta$ hyper-parameter in $\beta$-VAE Higgins et al. (2017).

The VAEs equipped with MIPE-transformation (MIPET-VAEs) can be trained with the following loss:

$$\mathcal{L}(\phi, \theta, \psi_{i \in [1,k]}; \boldsymbol{x}) = \mathcal{L}_{rec} - \mathcal{L}_{kl} - \mathcal{L}_{el} - \mathcal{L}_{cali}. \tag{37}$$

# D   Details of Experimental Environment

## D.1   Model Architecture

Table 7: VAE architecture for dSprites dataset.

| Encoder | Decoder |
|---------|---------|
| Input $64 \times 64$ binary image | input $\in \mathbb{R}^{10}$ |
| $4 \times 4$ conv. 32 ReLU. stride 2 | FC. 128 ReLU. |
| $4 \times 4$ conv. 32 ReLU. stride 2 | FC. $4 \times 4 \times 64$ ReLU. |
| $4 \times 4$ conv. 64 ReLU. stride 2 | $4 \times 4$ upconv. 64 ReLU. stride 2. |
| $4 \times 4$ conv. 64 ReLU. stride 2 | $4 \times 4$ upconv. 32 ReLU. stride 2. |
| FC. 128. FC. $2 \times 10$ | $4 \times 4$ upconv. 32 ReLU. stride 2. |
| | $4 \times 4$ upconv. 1. stride 2 |

Table 8: VAE architecture for 3D Shapes, and 3D Cars datasets. For exceptional case, CLG-VAE, we ues ten dimension size on 3D Shapes dataset Zhu et al. (2021).

| Encoder | Decoder |
|---------|---------|
| Input $64 \times 64 \times 3$ RGB image | input $\in \mathbb{R}^6$ (3D Shapes), $\mathbb{R}^{10}$ (3D Cars) |
| $4 \times 4$ conv. 32 ReLU. stride 2 | FC. 256 ReLU. |
| $4 \times 4$ conv. 32 ReLU. stride 2 | FC. $4 \times 4 \times 64$ ReLU. |
| $4 \times 4$ conv. 64 ReLU. stride 2 | $4 \times 4$ upconv. 64 ReLU. stride 2. |
| $4 \times 4$ conv. 64 ReLU. stride 2 | $4 \times 4$ upconv. 32 ReLU. stride 2. |
| FC. 256. FC. $2 \times 10$ | $4 \times 4$ upconv. 32 ReLU. stride 2. |
| | $4 \times 4$ upconv. 3. stride 2 |

## D.2   Details of Setting

Table 9: Hyper-parameters for dSprites, 3D Cars, and 3D Shapes. The epochs for dSprites and 3D cars are 30 and 200, respectively. lr is learning rate, latent dim is latent vector size, group reconst is group reconstruction, and forward group is forward group pass.

(a) Hyper-parameters for dSprites and 3D Cars

| models | hyper-parameters | values |
|--------|------------------|--------|
| common | batch size | 256 |
| | epoch | $\{30, 200\}$ |
| | optim | Adam |
| | lr | 4e-4 |
| | lr for MIPET | 4e-4 |
| | weight decay | 1e-4 |
| | latent dim | 10 |
| $\beta$-VAE | # of IE and EF | $\{1, 2, 4, 10\}$ |
| $\beta$-TCVAE | $\beta$ | $\{4, 6\}$ |
| | # of IE and EF | $\{1, 3\}$ |
| | $\alpha, \gamma$ | 1.0 |
| CLG-VAE | $\lambda_{decomp}$ | 40 |
| | $\lambda_{hessian}$ | 40 |
| | forward group | 0.2 |
| | group reconst | $\{0.2, 0.5, 0.7\}$ |

(b) Hyper-parameters for 3D Shapes

| models | hyper-parameters | values |
|--------|------------------|--------|
| common | batch size | 256 |
| | epoch | 67 |
| | optim | Adam |
| | lr | 4e-4 |
| | lr for MIPET | 4e-4 |
| $\beta$-VAE | # of IE and EF | $\{1, 2, 4, 10\}$ |
| | weight decay | 0.0 |
| | latent dim | 6 |
| $\beta$-TCVAE | $\beta$ | $\{4, 6\}$ |
| | # of IE and EF | $\{1, 3\}$ |
| | $\alpha, \gamma$ | 1.0 |
| | weight decay | 1e-4 |
| | latent dim | 6 |
| CLG-VAE | $\lambda_{decomp}$ | 40 |
| | $\lambda_{hessian}$ | 40 |
| | forward group | 0.2 |
| | group reconst | $\{0.2, 0.5, 0.7\}$ |
| | weight decay | 0.0 |
| | latent dim | 10 |

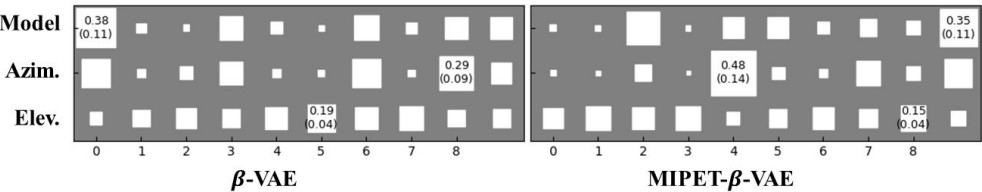

Figure 8: 3D Car Dataset: Azim. refers to the azimuth factor, and Elev. refers to the elevation factor.

## D.3 Additional Result of Relation Between Factors and Latent Vector Dimensions

In the 3D Car dataset, a trade-off was observed. While the maximum value across models and evaluation factors showed a slight decrease, the azimuth factor exhibited a notable improvement compared to the declines seen in other factors.

## D.4 Impact of Symmetric Matrix Exponential

Table 10: The ratio of seeds to show better performance with symmetric matrix

| dSprites | 3D Shapes | 3D Cars |
|----------|-----------|---------|
| 0.58     | 0.56      | 0.67    |

We empirically show the benefit of using a symmetric matrix for $\psi$. Table 10 shows the ratio of runs with a symmetric matrix, which shows better performance than unrestricted matrices, to the total 240 (60 models $\times$ 4 metrics) runs for each dataset. All results are higher than 0.5, which implies that the constraint enhances I2L equivariance even with uncertain factors.

## D.5 Additional Experiment of Computing Complexity

Table 11: Training complexity.

| # of IE | Complexity |
|---------|------------|
| 0       | $\times$ 1.00 |
| 1       | $\times$ 0.75 |
| 3       | $\times$ 0.50 |
| 4       | $\times$ 0.33 |

We additionally estimate the computing complexity depending on the number of IPE-transformation. The results are in Table 11 and represent the training time complexity compare to baselines (when the number of IE is equal to 0).

## D.6 Semantic Mask: Mapping Gaussian Distribution to Diverse Exponential Family

As shown in Fig. 9, MIPET-VAE with semantic mask preserves the dataset structure compared to VAE. Also, EF conversion with semantic mask maps the Gaussian distribution to the exponential family compared to MIPET without the semantic mask and VAE cases.

## D.7 Qualitative Analysis

We randomly sample an image for each dimension of the latent vector space and creates 10 variants of its generated latent vector by selecting values from {-2, 2} with 10 intervals for the dimension, then generate their corresponding output images. For the generation, we select $\beta$-TCVAE (6), which shows the best FVM

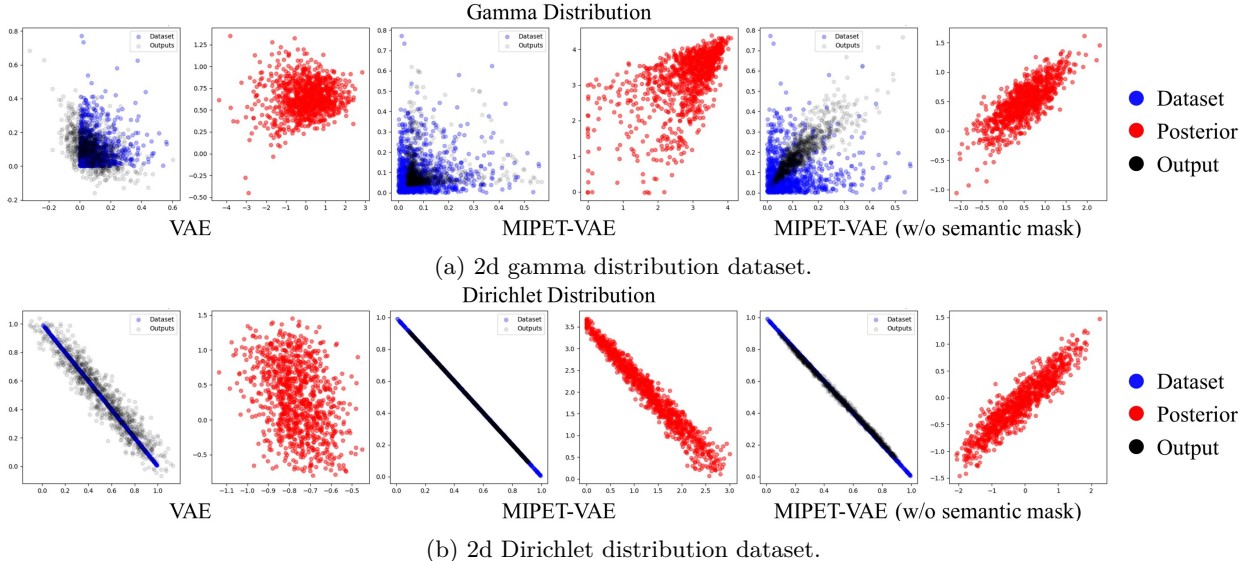

(a) 2d gamma distribution dataset.

(b) 2d Dirichlet distribution dataset.

Figure 9: 2d exponential family distribution dataset.

scores in dSprites dataset. Thereafter, we evaluate the semantic roles of each dimension before and after applying MIPE-transformation function.

In Figure 10, $\beta$-TCVAE struggles with y-position and rotation, as shown on the $6^{th}$ row, and with scale and shape represented on the $7^{th}$ row. On the contrary, MIPET-$\beta$-TCVAE separates y-position and rotation factor ($10^{th}$, and $7^{th}$ rows), also the activated dimensions of MIPET-$\beta$-TCVAE are not overlapped with each factor. Applied our method on $\beta$-TCVAE shows better disentangled representation on dSprites dataset. These results also show that our proposed method improves disentangled representation learning. As shown in the Figure 11, $\beta$-VAE struggles with rotation and scale factors in $4^{th}$ dimension. Also, it struggles with x-position and scale factors in $8^{th}$ dimension, and x-position and rotation factors in $9^{th}$ dimension. However, MIPET-$\beta$-VAE only struggles with rotation and shape factors in $5^{th}$ dimension. As shown in the Figure 12, CLG-VAE struggles with rotation and shape factors in $2^{nd}$ dimension, and shape and scale factors in $7^{th}$ dimension. However, MIPET-CLG-VAE separates rotation and shape factors in $10^{th}$, and $1^{st}$ dimensions respectively.

The qualitative analysis with 3D Shapes dataset, as shown in the Figure 13, $\beta$-TCVAE struggles with shape and scale in $5^{th}$ dimension. However MIPET-$\beta$-TCVAE As shown in the Figure 14, CLG-VAE struggles with shape and wall color factors in $4^{th}$ dimension, and shape and object color factors in $7^{th}$ dimension. In particular, it struggles with tree factors in $9^{th}$ dimension. On the other hand, MIPET-CLG-VAE separates shape, wall, and object color factors.

The qualitative analysis with 3D Cars dataset, as show in Figure 15, the left side is the $\beta$-TCVAE result, and it struggles with body, and azimuth factors shown in the $7^{th}$ row. However, MIPET-$\beta$-TCVAE separates azimuth ($6^{th}$ row) and body ($1^{st}$ row). In particular, MIPET-$\beta$-TCVAE learns *color* factor ($3^{rd}$ row) which does not exist on $\beta$-TCVAE.

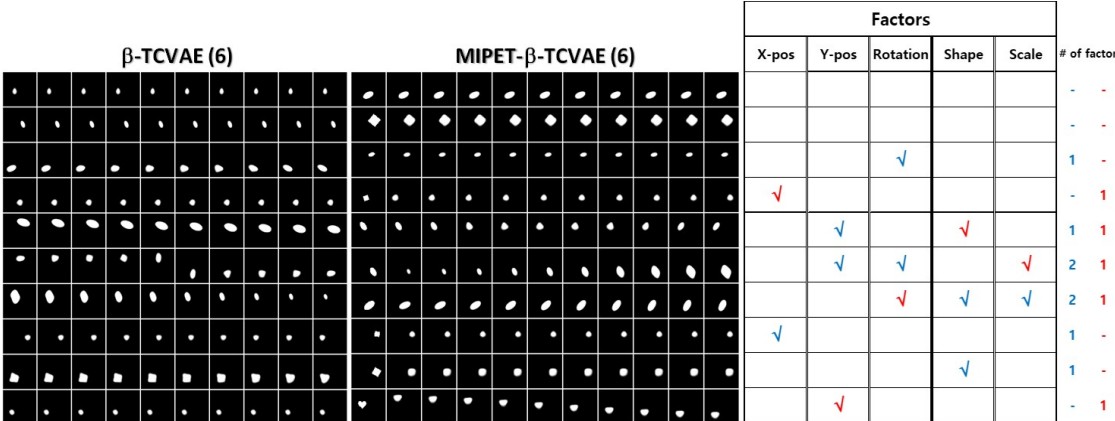

Figure 10: Qualitative results on dSprites. The left-side grids are input images and their variants by changing activations of each dimension of latent vectors. The first row shows input images. The right-side table shows matching pre-defined factors of the dataset (red: MIPET, blue: no MIPET).

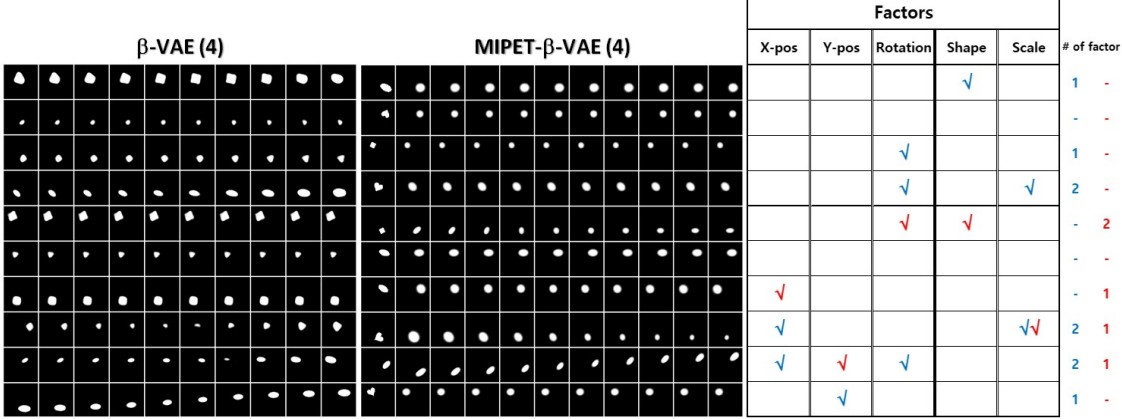

Figure 11: Qualitative analysis result of $\beta$-VAE and MIPET-$\beta$-VAE.

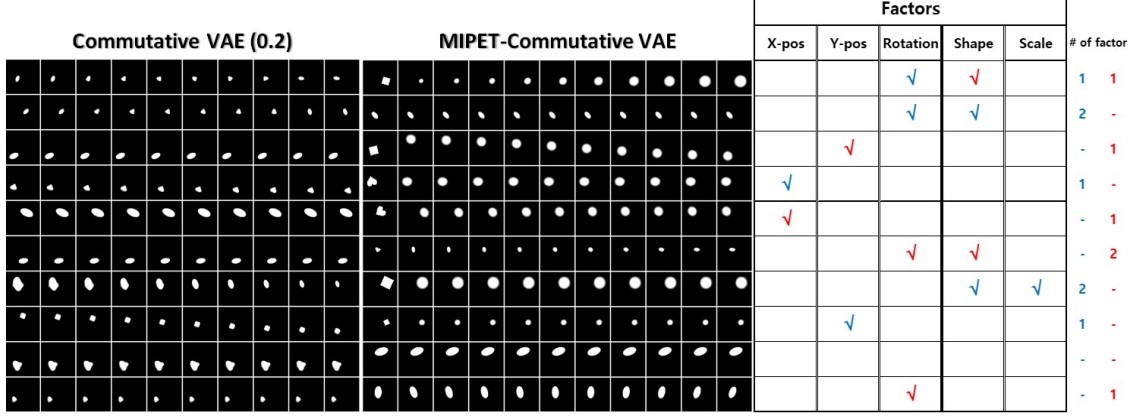

Figure 12: Qualitative analysis result of CLG-VAE (0.2) and MIPET-CLG-VAE (0.2) with dSprites.

**β-TCVAE (4)**  **MIET-β-TCVAE (4)**

| | Factors | | | | | | # factor | |
|---|---|---|---|---|---|---|---|---|
| Shape | Orien | Scale | Wall | Floor | Object | | | |
| | √ | | | | | √ | 1 | 1 |
| √√ | | | | | | | 1 | 1 |
| | | | √√ | | | | 1 | 1 |
| | | | | √√ | | | 1 | 1 |
| √ | | √√ | | | | | 2 | 1 |
| | √ | √ | | | | √√ | 2 | 2 |

Figure 13: The Shape is object shape, Orien is an orientation of object, Scale is a scale factor of object, Wall is wall color factor, Floor is floor color, and Object is object color factors. It represents the $\beta$-VAE ($\beta = 2$) results.

**Commutative VAE (0.2)**  **MIPET-Commutative VAE (0.2)**

| | Factors | | | | | | # of factor | |
|---|---|---|---|---|---|---|---|---|
| Shape | Orien | Scale | Wall | Floor | Object | | | |
| | √ | | | | | | - | 1 |
| | √ | | | | | | 1 | - |
| | | | | | | | - | - |
| √ | | | √ | | | | 2 | - |
| | | | | | | | - | - |
| | | | | | √ | | - | 1 |
| √ | | | | √ | √ | | 2 | 1 |
| √ | | √ | | √ | | | 1 | 2 |
| √ | | √ | √ | | √ | | 3 | 1 |
| | | | | | | | - | - |

Figure 14: Qualitative analysis result of CLG-VAE (0.2) and MIPET-CLG-VAE (0.2) with 3D Shapes.

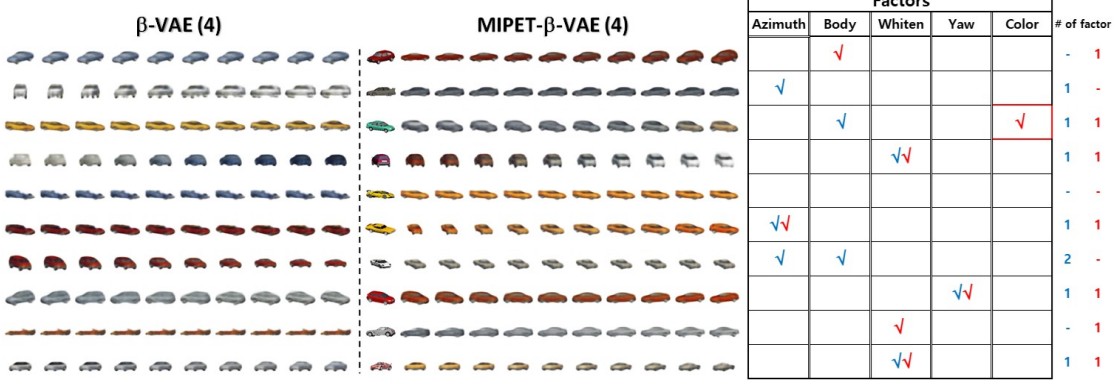

**β-VAE (4)**  **MIPET-β-VAE (4)**

| | Factors | | | | | # of factor | |
|---|---|---|---|---|---|---|---|
| Azimuth | Body | Whiten | Yaw | Color | | | |
| | √ | | | | | - | 1 |
| √ | | | | | | 1 | - |
| | √ | | | √ | | 1 | 1 |
| | | √√ | | | | 1 | 1 |
| | | | | | | - | - |
| √√ | | | | | | 1 | 1 |
| √ | √ | | | | | 2 | - |
| | | | √√ | | | 1 | 1 |
| | | √ | | | | - | 1 |
| | | √√ | | | | 1 | 1 |

Figure 15: Qualitative analysis result of $\beta$-VAE (4.0) with 3D Cars.

