# OpenReview forum: "Multiple Invertible and Partial-Equivariant Function for Latent Vector Transformation to Enhance Disentanglement in VAEs"
_TMLR — Rejected by TMLR_

### Review · Reviewer_4YMF · 2025-03-01

**Summary Of Contributions:**

The submission imparts additional structure to a VAE’s latent space in order to promote disentanglement.  The additional modules can be inserted immediately after a standard VAE’s latent space and before the decoder, and include highly structured latent-to-latent transformations, new regularizations based off of a connection with exponential family distributions, and a method of semantic masking.

**Audience:**

Yes

**Claims And Evidence:**

No

**Requested Changes:**

I think the changes required for acceptance are too significant at this stage.  Below are some minor points that could strengthen a future revision.

- Please use \citep{} for the references that are meant to be parenthetical.
- The paper talks of “unrestricted distributions” of $\hat{z}$, yet if $z$ follows a Gaussian distribution and $\phi(\cdot)$ is just a linear transformation, shouldn’t $\hat{z} = \phi(z)$ also be a Gaussian distribution?
- What does Fig 3 add that isn’t in Fig 2?
- It seems the natural parameter generator (NPG) work is not cited correctly, and Charpentier et al. (2022) is doubly cited.
- Fig 4, MIPET, row 5 seems to have rotation information as well?
- In Sec 3.2.3, it seems “probability density function” is meant instead of “power density function”?
- In the code for `Invert_Equiv_Func` in `ie_transformation.py`, $\phi$ is created as the product of two matrices, but this is not discussed in the paper.  Is the use of the product intentional/beneficial?

**Strengths And Weaknesses:**

## Strengths

The submission draws from several different ideas from unsupervised disentanglement and combines them into a framework with some intriguing elements.

## Weaknesses
I found the paper and the method very difficult to understand.  The quality of the written text is often poor, and the method itself is convoluted.  The various pieces of motivation from prior work are disconnected and often flawed, leading to a confusing patchwork of methodological choices.

First and foremost, the central motivation of leveraging invertibility and (partial) equivariance refers to the latent-to-latent transformation and is therefore far removed from the data itself.
The transformation is parameterized as an exponentiated symmetric matrix after the latent space, and its invertibility and elevated-chance-of-equivariance are argued to impart a desirable structure to the latent space.  The motivation for invertibility is a loose connection with InteL-VAEs from Miao et al. (2022), except that the latter imposed nontrivial topological structure on the latent space as an inductive bias.  The motivation for equivariance is taken from Higgins et al. (2018), except that the latter focused on equivariance in the transformation from data to latents, not from one latent space to another.

Then there is a connection with Peebles et al. (2020) where entanglement was encouraged through driving the off diagonal terms of the Hessian **from latent space to the reconstructed output** to zero.  The authors claim that the Hessian of the latent-to-latent transformation being (trivially) zero encourages disentanglement, but this has no connection with the data.  One can apply any arbitrary linear transformation to a latent space before subsequent decoding, leading to a Hessian matrix of zero for the transformation between latent spaces, and there would be no reason to expect that an inductive bias was inserted nor the properties of the latent space changed.

As far as I can tell from the paper and the attached code, the exponential family (EF) conversion is completely detached from the decoding process, meaning that the involved KL calculations are also untethered from the generative process.  They might provide some implicit regularization, but at the cost of tons of additional complexity.

The authors claim a connection to beta-VAEs, which is also superficial.  The arbitrary way multiple IPE module losses are combined leads to a relative weighting between the KL terms and the reconstruction loss equal to k (the number of terms, Eqn 10), but the particular weighting is not grounded in anything and does not allow any control over reweighting, which is central to a beta-VAE.

Finally, the motivation behind having multiple IPE transformations is dubious -- the authors write in Sec. 3.3 that “each IPE-transformation function operates independently”, but that’s not true if they all share the same decoder.  I imagine that it should create pressure for the transformations to become the same, effectively turning MIPE to IPE with a relative weighting between reconstruction and KL being different than unity.

---

> ### Author Response · Authors · 2025-04-07
> **Response to Reviewer 4YMF 1**
>
> ### Summary of Motivations
> > We first briefly summarize our motivations to show the connections with previous works, and we add it at the Introduction section in the revised version. And we highlight it.
> >
> > $\\textbf{Need for the Exponential Family.}$
> [1] demonstrates that regularizing the aggregate posterior, rather than the posterior fitted to a complex prior, influences disentanglement learning. It shows the necessity of a **complex prior** for disentanglement learning [2]. Motivated by these insights, we propose the **‘exponential conversion’** to transform a Gaussian normal distribution to an exponential family (a type of complex distribution).
> >
> > $\\textbf{Need for Equivariance.}$
> [3] emphasized the importance of inductive biases for disentanglement. [4] further reframed disentangled representation learning from the viewpoint of symmetry. Building on this, subsequent work [5] has shown that **equivariant** architectures improve disentanglement performance, providing meaningful structure in the latent space.
> However, naively applying **exponential conversion** does not guarantee preservation of partial equivariance. Therefore, we implemented the EF-conversion via an exponential matrix to satisfy the conditions necessary for preserving **partial-equivariance**; specifically, the transformation must be 1) symmetric and 2) **invertible**. Then we mathematically show that **a symmetric matrix exponential** is an appropriate transformation to preserve partial-equivariance.
>
>
> > [1] Emile Mathieu, Tom Rainforth, N. Siddharth, and Y. Teh. Disentangling disentanglement in variational autoencoders. In International Conference on Machine Learning, 2018.
> >
> > [2] Ning Miao, Emile Mathieu, Siddharth N, Yee Whye Teh, and Tom Rainforth. On incorporating inductive biases into VAEs. In International Conference on Learning Representations, 2022.
> >
> > [3] Francesco Locatello, Stefan Bauer, Mario Lucic, Gunnar Raetsch, Sylvain Gelly, Bernhard Schölkopf, and Olivier Bachem. Challenging common assumptions in the unsupervised learning of disentangled representations. International Conference on Machine Learning.
> >
> > [4] Irina Higgins, David Amos, David Pfau, Sébastien Racanière, Loïc Matthey, Danilo J. Rezende, and Alexander Lerchner. Towards a definition of disentangled representations. CoRR, abs/1812.02230, 2018b.
> >
> > [5] Xinqi Zhu, Chang Xu, and Dacheng Tao. Commutative lie group vae for disentanglement learning. International Conference on Machine Learning, 2021.
>
> ---
> ### R1 C1.
> > The motivation for invertibility is a loose connection with InteL-VAEs from [2], except that the latter imposed nontrivial topological structure on the latent space as an inductive bias.
>
> ### R1 A1.
> > We acknowledge that our previous version did not sufficiently explain the relevance to the referenced works, which may have caused misunderstandings, and we have addressed this as follows. As mentioned in the Summary of Motivation, our reason for using an **invertible matrix** is to preserve the existing **partial equivariance** during the transformation into a complex distribution [1, 2]. Thus, our primary reason for citing [2] was to highlight that employing** complex distributions** can better capture the underlying data structure. We have revised the manuscript to reflect these points more clearly—please refer to the **highlighted parts** in sections ①, ②, ④–⑧, and ⑩–(13). Additionally, to better emphasize the importance of invertibility, we have **modified the order of the proofs**, changing the previous Proposition 3.1 to Proposition 3.3 in the revised version.
>
> ---
>
> ### R1 C2.
> > The motivation for equivariance is taken from [4], except that the latter focused on equivariance in the transformation from data to latents, not from one latent space to another.
>
> ### R1 A2.
> > Our main goal is also to ensure **equivariance between data and latent representations**, as emphasized by Higgins. In the process of transforming the Gaussian distribution into a diverse exponential family, the pre-existing partial-equivariance between the data and latent space is typically not preserved. Therefore, we propose a method that maximally ensures this partial-equivariance. Additionally, **Fig. 4 illustrates** that when applying the IPE transformation, **partial-equivariance is preserved from the dataset space to the transformed latent space**. Furthermore, **Propositions 3.1-3.3 show** that a symmetric matrix exponential is an appropriate matrix to **preserve partial-equivariance from dataset space to transformed latent vector space**. I have added content to Section 3.1 to clearly reflect the above points, and highlighted the additions for clarity (⑩-⑫ sentences in the revised version). Also we briefly explained the equivariance between the dataset space and transformed latent vector space in section 3.1.1.
> ---

---

> ### Author Response · Authors · 2025-04-07
> **Response to Reviewer 4YMF 2**
>
> ### R1 C3.
> > Then there is a connection with [6] where entanglement was encouraged through driving the off diagonal terms of the Hessian from latent space to the reconstructed output to zero. The authors claim that the Hessian of the latent-to-latent transformation being (trivially) zero encourages disentanglement, but this has no connection with the data. One can apply any arbitrary linear transformation to a latent space before subsequent decoding, leading to a Hessian matrix of zero for the transformation between latent spaces, and there would be no reason to expect that an inductive bias was inserted nor the properties of the latent space changed.
>
> ### R1 A3.
> > We intended to clearly explain why our method improves disentanglement learning; however, as you pointed out, the explanation provided could apply to any linear transformation, and we acknowledge that our description was misleading. Accordingly, we have removed that section in the revised version. While we may not yet provide a fully rigorous theoretical justification, we believe our model demonstrates improved performance compared to baseline methods. Furthermore, the necessity of the invertible matrix and EF conversion is empirically supported through ablation studies.
> [6] William Peebles, John Peebles, Jun-Yan Zhu, Alexei A. Efros, and Antonio Torralba. The hessian penalty: A weak prior for unsupervised disentanglement. In Proceedings of European Conference on Computer Vision (ECCV), 2020.
>
> ---
>
> ### R1 C4.
> > As far as I can tell from the paper and the attached code, the exponential family (EF) conversion is completely detached from the decoding process, meaning that the involved KL calculations are also untethered from the generative process. They might provide some implicit regularization, but at the cost of tons of additional complexity.
>
> ### R1 A4.
> > The EF conversion is **directly connected to the decoding process**. If your reference regarding the use of **'detach'** pertains to lines 185 and 186 in the **'ie_transformation.py' file** of our provided code, please note that this part is not utilized during the actual model training process. Consequently, the **latent vector z, newly transformed via the EF conversion, is fed into the decoder**. The latent vector z, together with the epsilon value, is then used to compute $\\mathcal{L}_{kl}, \\mathcal{L}_{el}$.
>
> ---
>
> ### R1 C5.
> > The authors claim a connection to beta-VAEs, which is also superficial. The arbitrary way multiple IPE module losses are combined leads to a relative weighting between the KL terms and the reconstruction loss equal to k (the number of terms, Eqn 10), but the particular weighting is not grounded in anything and does not allow any control over reweighting, which is central to a beta-VAE.
>
> ### R1 A5.
> > We **mathematically demonstrate the connection** between the number of IPE-transformations and the hyper-parameter $\\beta$ of $\\beta$-VAE **in Appendix C.1**. I have added the relevant content in the revised paper to establish a clear connection with the Appendix in section 3.3 and highlight it ((14) sentence).
>
> ---
>
> ### R1 C6.
> > Finally, the motivation behind having multiple IPE transformations is dubious -- the authors write in Sec. 3.3 that “each IPE-transformation function operates independently”, but that’s not true if they all share the same decoder. I imagine that it should create pressure for the transformations to become the same, effectively turning MIPE to IPE with a relative weighting between reconstruction and KL being different than unity.
>
> ### R1 A6.
> > Each IPE-transformation **operates independently**, even though they share the same decoder. We conducted an **additional simple experiment** to illustrate how the outputs of different IPE-transformations are represented in the latent vector space. **As shown in Figure 6**, the results of different IPE-transformations clearly represent distinct distributions. We add experimental results in section 5.2.2 in the revised version.
>
> ---
>
> ### R1 Q1.
> > Please use \citep{} for the references that are meant to be parenthetical.
>
> ### R1 QA1.
> > We revise it in the revised version paper.
>
> ---
>
> ### R1 Q2.
> > The paper talks of “unrestricted distributions” of $\\hat{z}$, yet if z follows a Gaussian distribution and $\\phi(\\cdot)$ is just a linear transformation, shouldn’t $\\hat{z}=\\phi(z)$ also be a Gaussian distribution?
>
> ### R1 QA2.
> > That's correct. **However, we address this issue using an implicit semantic mask**. Without applying the implicit semantic mask, our method indeed approaches the Gaussian distribution, as you pointed out. However, when the implicit semantic mask is applied, our experiments demonstrate that it becomes closer to an exponential family distribution. Furthermore, **Figure 5 shows** that our method can approximate exponential family distributions, including beta distribution. **We add this experimental result in section 5.2.1** in the revised version.
>
> ---

---

> > ### Comment · Reviewer_4YMF · 2025-04-12
> >
> > R1 A4: No, I am looking at line 33 in `ie_transformation.py`.  The L2L transformation is applied to the latent vector (`latent_z` becomes `new_z`) before handing it to the decoder, but the EF conversion doesn't change the latent vector at all.  It is a convoluted regularization term that has no direct effect on the decoding.  Please correct me if you think I'm misunderstanding.

---

> > > ### Author Response · Authors · 2025-04-14
> > > **Response to Reviewer 4YMF 4**
> > >
> > > > Yes, that is correct. The EF conversion is not a module that directly transforms $\hat{z}$ into an exponential family form before decoding, but **rather a regularization term** (as you mentioned at R1 W5) that guides $\hat{z}$ toward the exponential family through the L2L transformation, which is updated via $\mathcal{L}_{el}$, and $\mathcal{L}_{kl}$ in the previous iteration.
> > >
> > > > We agree with your point, because the L2L transformation did not transform the $z$ to the exponential family at the current iteration. However, **it is difficult to assert** that this process 1) provide **implicit** regularization (as you mentioned at R1 W5) or 2) does **not directly** affect decoding. Since the L2L transformation is iteratively learned through the EF conversion loss over multiple iterations, it is reasonable to assume that, once the **objective loss has saturated**, the decoder would also be directly affected. Also, as shown in additional experiments (Figure 5), this regularization term **led to** distributions that more **closely approximated the exponential family**, suggesting that the EF conversion indeed influences such approximation.
> > >
> > > > In the revised version of the paper, the EF conversion component in Figure 1 is depicted as though it directly transforms $\hat{z}$, which may obscure the actual mechanism we propose. As other reviewers are currently evaluating the paper, we believe that revising it mid-review would be difficult. Therefore, we plan to revise Figure 1 after the review process has been completed.

---

> ### Author Response · Authors · 2025-04-07
> **Response to Reviewer 4YMF 3**
>
> ### R1 Q3.
> > What does Fig 3 add that isn’t in Fig 2?
>
> ### R1 QA3.
> > Fig. 2 in the revised version demonstrates that our method preserves partial equivariance. Additionally, Fig. 3 illustrates that partial equivariance is maintained from the dataset space to the transformed latent space when employing the IPE transformation.
>
> ---
>
> ### R1 Q4.
> > It seems the natural parameter generator (NPG) work is not cited correctly, and [7] is doubly cited.
>
> ### R1 QA4.
> > We did not directly use the method proposed by [7]. Instead, we implemented a simple MLP structure to represent the exponential family parameter space derived from the latent space, as described in that paper. Additionally, we have corrected the duplicate citation of the same paper.
> [7] Bertrand Charpentier, Oliver Borchert, Daniel Zugner, Simon Geisler, and Stephan Gunnemann. Natural posterior network: Deep bayesian predictive uncertainty for exponential family distributions. In International Conference on Learning Representations, 2022.
> ---
>
> ### R1 Q5.
> > Fig 4, MIPET, row 5 seems to have rotation information as well?
>
> ### R1 QA5.
> > Yes, we have revised this part. Although our method involves changes in two factors in the 5th dimension, the baseline shows overlapping factors in the 6th and 7th dimensions. Therefore, our method still demonstrates improvements over the baseline.
>
> ---
>
> ### R1 Q6.
> > In Sec 3.2.3, it seems “probability density function” is meant instead of “power density function”?
>
> ### R1 QA6.
> > Yes, you are correct. The aforementioned content has been revised.
>
> ---
>
> ### R1 Q7.
> > In the code for Invert_Equiv_Func in ie_transformation.py, $\\phi$ is created as the product of two matrices, but this is not discussed in the paper. Is the use of the product intentional/beneficial?
>
> ### R1 QA7.
> > We multiplied two invertible matrices to increase the model's capacity. Since the product of two invertible matrices remains invertible, we employed the multiplication of two matrices.
>
> ---

---

### Review · Reviewer_XTxf · 2025-03-02

**Summary Of Contributions:**

This paper focuses on improving the VAE-based disentangled representation learning by improving the usage of inductive bias. This paper proposes several plug-in functions and losses that can fit into most of the current SOTA VAE modules to improve performance. Specifically, MIPE transformation enforces the invertibility and partial-equivariant of latent-to-latent transformation, and IPE-transformation builds Gaussian distribution from any arbitrary distribution. Empirical results validate the advantage of the proposed method with several VAE-based models.

**Audience:**

Yes

**Claims And Evidence:**

Yes

**Requested Changes:**

Suggestions:
Please refer to the weakness.

**Strengths And Weaknesses:**

Strength:
The paper identified some critical challenges in improving the VAE-based disentangled representation learning model. The proposed function is promising in combining existing VAE models to resolve the identified challenges. The theoretical analysis seems solid, and the empirical results look good. I appreciate Figures 1 to 3, which help me understand the motivation and theory behind the proposed method.

Weakness:
However, in general, I find this paper very hard to read and follow. Objectively speaking, the proposed method (especially the theoretical analysis) is quite complicated, so it can be challenging to explain everything clearly. But this paper definitely has a lot of room for improvement. For example, one of the core motivations behind the paper is that "the actual implementation of such bias is still vague" (as written in the abstract). Then, such bias can be effectively implemented by the proposed method (MIPE or IPE) in detail, which is not clear to me. Actually, Section 3 barely mentions "bias." Similarly, although the introduction section and some analysis in section 3 give me some idea that "invertible" and "equivariant" are important to disentangled representation learning, it is not very straightforward to me the detailed reason behind it and how the proposed method resolves it in simple words. I appreciate the detailed theoretical analysis and proofs, but adding more simple motivation and high-level explanations can better prepare the reader before diving into detailed theories. Also, Locatello et al. (2019) have already proved that disentangled representation learning can not be done in an unsupervised fashion (which is also pointed out by the paper), so I don't think it is necessary to include so many discussions on unsupervised learning. When reading those discussions, I wondered if the paper should propose an "improved" unsupervised method. Also, there are so many typos, grammar errors, unclear expressions, etc, that need to be polished. For example, there are two obvious typos, "tra nsformation" and "familyt" in the paragraph before the "Method" section that should be spotted by proofreading.

---

> ### Author Response · Authors · 2025-04-07
> **Response to Reviewer XTxf 1**
>
> ### Summary of Motivations
> > We first briefly summarize our motivations to show the connections with previous works, and we add it at the Introduction section in the revised version. And we highlight it.
> >
> > $\\textbf{Need for the Exponential Family.}$
> [1] demonstrates that regularizing the aggregate posterior, rather than the posterior fitted to a complex prior, influences disentanglement learning. It shows the necessity of a **complex prior** for disentanglement learning [2]. Motivated by these insights, we propose the **‘exponential conversion’** to transform a Gaussian normal distribution to an exponential family (a type of complex distribution).
> >
> > $\\textbf{Need for Equivariance.}$
> [3] emphasized the importance of inductive biases for disentanglement. [4] further reframed disentangled representation learning from the viewpoint of symmetry. Building on this, subsequent work [5] has shown that **equivariant** architectures improve disentanglement performance, providing meaningful structure in the latent space.
> However, naively applying **exponential conversion** does not guarantee preservation of partial equivariance. Therefore, we implemented the EF-conversion via an exponential matrix to satisfy the conditions necessary for preserving **partial-equivariance**; specifically, the transformation must be 1) symmetric and 2) **invertible**. Then we mathematically show that **a symmetric matrix exponential** is an appropriate transformation to preserve partial-equivariance.
>
>
> > [1] Emile Mathieu, Tom Rainforth, N. Siddharth, and Y. Teh. Disentangling disentanglement in variational autoencoders. In International Conference on Machine Learning, 2018.
> >
> > [2] Ning Miao, Emile Mathieu, Siddharth N, Yee Whye Teh, and Tom Rainforth. On incorporating inductive biases into VAEs. In International Conference on Learning Representations, 2022.
> >
> > [3] Francesco Locatello, Stefan Bauer, Mario Lucic, Gunnar Raetsch, Sylvain Gelly, Bernhard Schölkopf, and Olivier Bachem. Challenging common assumptions in the unsupervised learning of disentangled representations. International Conference on Machine Learning.
> >
> > [4] Irina Higgins, David Amos, David Pfau, Sébastien Racanière, Loïc Matthey, Danilo J. Rezende, and Alexander Lerchner. Towards a definition of disentangled representations. CoRR, abs/1812.02230, 2018b.
> >
> > [5] Xinqi Zhu, Chang Xu, and Dacheng Tao. Commutative lie group vae for disentanglement learning. International Conference on Machine Learning, 2021.
>
> ---
>
> ### R2 C1.
> > For example, one of the core motivations behind the paper is that "the actual implementation of such bias is still vague" (as written in the abstract). Then, such bias can be effectively implemented by the proposed method (MIPE or IPE) in detail, which is not clear to me. Actually, Section 3 barely mentions "bias."
>
> ### R2 A1.
> > Although **previous studies have also not explicitly defined what inductive bias is**, they have shown in [4, 5] that leveraging equivariant properties serves as an effective form of inductive bias for disentanglement learning. **Based on this insight**, we also injected inductive bias by preserving equivariant properties—**an implicitly expressed** but practically beneficial bias for disentanglement learning. We have revised the manuscript to better highlight this point. Please refer to the **highlighted sentences ③ and ⑨ in the revised version**.
>
> ---
>
> ### R2 C2.
> > Similarly, although the introduction section and some analysis in section 3 give me some idea that "invertible" and "equivariant" are important to disentangled representation learning, it is not very straightforward to me the detailed reason behind it and how the proposed method resolves it in simple words.
>
> ### R2 A2.
> > As mentioned in the **Summary of Motivation**, we employed an **invertible matrix to preserve the existing partial-equivariance** during the transformation from a Gaussian distribution to a complex distribution [1, 2]. We have revised the manuscript to clearly reflect this point; please refer to the **highlighted sections ①, ②, ④–⑧, and ⑩–⑬**. Additionally, we modified only the **order of the theoretical proofs**—specifically, Proposition 3.1 in the original version has been revised to Proposition 3.3—to more clearly emphasize the importance of invertibility.
>
> ---

---

> ### Author Response · Authors · 2025-04-07
> **Response to Reviewer XTxf 2**
>
> ### R2 C3.
> > I appreciate the detailed theoretical analysis and proofs, but adding more simple motivation and high-level explanations can better prepare the reader before diving into detailed theories.
>
> ### R2 A3.
> > We mathematically demonstrate our method for incorporating the **advantages of the equivariant property as an inductive bias**. Specifically, **we prove why invertibility and partial-equivariance are necessary to preserve partial-equivariance**. First, we show that a **symmetric and invertible matrix (transformation) is always equivariant** with respect to a specific group ($G_S$). Second, we demonstrate that such a matrix constitutes **a partial-equivariant function** between the input dataset and the transformed latent vector space. Finally, we provide a proof explaining why the **symmetric and invertible matrix** is the most **appropriate transformation** in this context. To better highlight these points, we revised Section 3.1.2 in the updated version of the manuscript. **Please refer to the highlighted sentences ⑩, ⑪, and ⑫**.
>
> ---
>
> ### R2 C4.
> > Also, [3] have already proved that disentangled representation learning can not be done in an unsupervised fashion (which is also pointed out by the paper), so I don't think it is necessary to include so many discussions on unsupervised learning. When reading those discussions, I wondered if the paper should propose an "improved" unsupervised method.
>
> ### R2 A4.
> > What [3] demonstrated is that unsupervised learning is not possible **without inductive bias**. In other words, their study **highlights the necessity of inductive bias in the context of unsupervised learning**. In addition, previous work [5] introduced inductive bias by utilizing the property of **equivariance**, which proved effective in disentanglement learning. Based on these insights, **our study also focuses on how to effectively incorporate inductive bias**, and we address this by proposing a method that aims to preserve **partial equivariance** as much as possible. We have revised the manuscript to better highlight this point (please refer to highlighted sentences (0) and ③).
>
> ---
>
> ### R2 C5.
> > Also, there are so many typos, grammar errors, unclear expressions, etc, that need to be polished. For example, there are two obvious typos, "tra nsformation" and "familyt" in the paragraph before the "Method" section that should be spotted by proofreading.
>
> ### R2 A5.
> > We revised these points in the revised version.
>
> ---

---

### Review · Reviewer_KzYv · 2025-03-27

**Summary Of Contributions:**

This paper introduces MIPE-transformation, a new method to improve how Variational Autoencoders (VAEs) learn disentangled representations — where each dimension of the latent vector ideally corresponds to a separate semantic factor (like color, shape, or orientation). The approach has three key innovations:

* Invertible and Partially Equivariant (IPE) Transformations:
The authors design a special transformation applied to the latent vector that is both invertible (information-preserving) and sensitive to certain kinds of structured changes in the data (like rotation or scaling). This helps separate semantic features across different dimensions of the latent space.

* Exponential Family Projection (EF-conversion):
Since the transformation changes the latent distribution in ways that may no longer match the usual Gaussian assumption in VAEs, the authors introduce a projection that converts the latent vector into a more general family of probability distributions. This makes it easier to train the model effectively.

* Stacked MIPE Architecture:
The method stacks multiple IPE and EF blocks to build up stronger structure in the latent space. Each layer progressively helps isolate different semantic factors. This replaces the need for aggressive regularization (like in beta-VAE) with a more principled architectural design.

The authors conduct extensive experiments on three popular benchmark datasets (dSprites, 3D Shapes, and 3D Cars), and show that MIPE consistently improves over standard VAEs and other unsupervised disentanglement baselines. Visualizations and quantitative metrics confirm that MIPE leads to more interpretable latent representations. The paper also includes thorough ablation studies to justify each component.

**Audience:**

Yes

**Broader Impact Concerns:**

potential broader implications emerge in downstream applications:

* If extended to real-world data (e.g., facial images, surveillance video, or health records), improvements in disentangled representations could enable more powerful and controllable generative models, which may raise issues of privacy, identity manipulation, or misuse in deepfakes.

* If applied to socially sensitive factors (e.g., race, gender, emotion), automated disentanglement may inadvertently reinforce or expose biases present in training data — especially in the absence of strong supervision or fairness constraints.

* The use of stacked architectural priors such as MIPE may also influence how future generative models are interpreted or audited, particularly in applications requiring transparency or explainability (e.g., legal or clinical contexts).

**Claims And Evidence:**

Yes

**Requested Changes:**

* Add results on more complex datasets (recommended, not critical)
Include an experiment on datasets like CelebA or real-world colored objects (e.g., CUB) to demonstrate generalizability. Even if performance degrades, it would help illustrate the method’s boundaries.

* Explicit analysis of factor-to-latent alignment
While the paper shows latent traversals, it would strengthen the argument to add a quantitative or structured analysis on which IPE modules are disentangling which semantic factor (e.g., color, shape).

* Compare with more recent methods
Include baselines or comparisons to recent methods like TCVAE, FactorVAE, or unsupervised contrastive disentanglement approaches to better contextualize MIPE’s contribution.

* Discuss applicability to multimodal settings (optional)
The method seems extensible to vision-language domains (e.g., VAE+CLIP). A brief discussion in the conclusion on this possibility would enhance relevance.

**Strengths And Weaknesses:**

Strengths:

* The proposed IPE-transformation combines theoretical elegance (invertibility + symmetry-induced equivariance) with practical benefits for disentanglement. The use of matrix exponential transformations is mathematically principled and novel in the VAE context.

* The exponential family projection is a clever solution to the non-Gaussian posterior problem. It enables optimization without violating tractability or introducing excessive approximations.

* The MIPE architecture introduces a scalable and interpretable way of introducing inductive bias. It mirrors the success of deep stacking in neural networks but applies it to structured latent manipulation.

* Clear improvements over baseline and ablation models across all datasets and disentanglement metrics. Qualitative latent traversals also support the quantitative claims.

* Solid theoretical insights: The analysis in Sections 3.1.2–3.1.3 provides a formal rationale for using symmetric matrices and the benefits of equivariant mappings.

Weaknesses:

* Limited scalability to complex domains: The method is validated only on canonical toy datasets (dSprites, 3D Shapes, Cars). It remains unclear how the framework would perform in more complex or less structured real-world datasets, e.g., CelebA, ImageNet, or multi-modal domains.

* While the paper emphasizes interpretability via symmetry and stacking, these structures may become too rigid or suboptimal in settings with richer or entangled semantic spaces.

* Disentanglement interpretability is implicit: Although the architecture encourages disentanglement, there is no explicit alignment of latent dimensions to known factors (e.g., using weak supervision or language). Interpretability remains qualitative and relative.

---

> ### Author Response · Authors · 2025-04-07
> **Response to Reviewer KzYv**
>
> ### R3 C1.
> > Add results on more complex datasets (recommended, not critical) Include an experiment on datasets like CelebA or real-world colored objects (e.g., CUB) to demonstrate generalizability. Even if performance degrades, it would help illustrate the method’s boundaries.
>
> ### R3 A1.
> > We add the celebA dataset results in section 5.2.3 and Figure 7. in the revised version.
>
> ---
>
> ### R3 C2.
> > Explicit analysis of factor-to-latent alignment While the paper shows latent traversals, it would strengthen the argument to add a quantitative or structured analysis on which IPE modules are disentangling which semantic factor (e.g., color, shape).
>
> ### R3 A2.
> > We visualized the **DCI matrix**, which quantitatively measures the relationship between factors and latent variables, and the results can be found **in Figure 4 of the revised version**. A sparse matrix represents the most ideal case, and the case is considered more ideal when each row has a higher maximum value and a larger standard deviation. As shown in Figure 4, **our model demonstrates improved performance** on the dSprites and 3D Shapes dataset. This finding has been highlighted in both Figure 4 and Section 5.1.1 of the revised version.
>
> ---
>
> ### R3 C3.
> > Compare with more recent methods Include baselines or comparisons to recent methods like TCVAE, FactorVAE, or unsupervised contrastive disentanglement approaches to better contextualize MIPE’s contribution.
>
> ### R3 A3.
> > The experimental results for $\beta$-TCVAE have already been demonstrated, and we **have additionally included results for Factor-VAE**. Overall, it can be observed that our model shows **improved performance**. These results are presented in **Table 2**.
> >\\begin{array}{c|c|c|c|c|c|c|c|c}
> \\hline
> & FVM & FVM & MIG & MIG & SAP & SAP & DCI & DCI \\\\
> \\hline
> & Factor-VAE & MIPET & Factor-VAE & MIPET & Factor-VAE & MIPET & Factor-VAE & MIPET \\\\
> \\hline
> dSprites & 67.78(\pm7.48) & \textbf{68.38}(\pm8.55) & 14.67(\pm10.40) & \textbf{19.31}(\pm13.46) & 2.35(\pm2.32) & \textbf{4.24}(\pm3.36) & 22.58(\pm8.50) & \textbf{30.18}(\pm12.99) \\\\
> 3D Shapes & 77.43(\pm10.71) & \textbf{78.34}(\pm8.35) & 24.23(\pm26.13) & \textbf{48.59}(\pm10.58) & 4.99(\pm4.46) & \textbf{9.84}(\pm2.73) & \textbf{68.68}(\pm12.92) & 60.23(\pm9.90) \\\\
> 3D Cars & 89.64(\pm2.62) & \textbf{93.66}(\pm3.83) & \textbf{10.63}(\pm1.30) & 9.43(\pm1.12) & 2.51(\pm0.37) & \textbf{3.28}(\pm0.83) & \textbf{26.27}(\pm3.17) & 23.13(\pm0.72) \\\\
> \\hline
> \\end{array}
> ---
>
> ### R3 C4.
> > Discuss applicability to multimodal settings (optional) The method seems extensible to vision-language domains (e.g., VAE+CLIP). A brief discussion in the conclusion on this possibility would enhance relevance.
>
> ### R3 A4.
> > **We add discussion of vision-language domains in the section 6** (highlighted 18 sentence). In the revised version and highlight it: ‘Although our model does not show sufficiently improved results on complex datasets such as CelebA or beyond, we anticipate that utilizing architectures like CLIP, which directly train symmetry (transformation) information through language and vision datasets, would enable more efficient injection of inductive bias compared to unsupervised learning approaches.’
>
> ---

---

### Author Response · Authors · 2025-04-07

Dear, all Reviewers, We appreciate your valuable and fruitful comments to improve this research. The abbreviation ‘R’ refers to a reviewer, ‘C’ refers to a reviewer’s comment, ‘A’ refers to an answer, and ‘QA’ refers to an answer of request.

---

### Decision · Action_Editor_iHDE · 2025-05-17

**Recommendation:** Reject

**Comment:**

While the paper presents an interesting approach to interpretable representation learning, I regretfully recommend rejection at this time. The manuscript would benefit from significant clarity improvements, as the theoretical framework currently lacks sufficient high-level explanations to make it accessible to readers.

The evaluation is primarily conducted on standard benchmark datasets (dSprites, 3D Shapes, Cars; CelebA only partially), and I would encourage the authors to demonstrate the method's applicability to more complex real-world datasets. The proposed interpretability approach through symmetry and stacking may face challenges in domains with more entangled semantic spaces, and the disentanglement approach could be strengthened by incorporating explicit alignment between latent dimensions and interpretable factors.

Although the abstract mentions addressing "vague bias implementation," this connection could be more clearly developed throughout the paper. I believe the manuscript shows promise but would benefit from a clearer presentation of the relationships between the theoretical contributions and their practical implementation. With substantial revision to improve clarity and better justify the methodological choices, this work could make a valuable contribution to the field.

**Audience:**

The paper would be of interest to TMLR's audience.

**Claims And Evidence:**

The paper makes the following three claims:
1. A symmetric matrix exponential as a latent-to-latent vector transformation function induces inductive bias in VAEs.
2. A new training procedure and loss functions for VAEs allow to learn an unknown latent vector distribution as an approximate exponential family.
3. A novel MIPE-transformation architecture is widely applicable to state-of-the-art VAEs.

The authors presented empirical results and also updated them during the rebuttal. However, as pointed out by some of the reviewers, the results on easy and simple-domain datasets are not convincing enough to support the claims. The framework's effectiveness remains untested on complex real-world datasets like CelebA (not fully) or ImageNet. Additionally, its emphasis on symmetry and stacking structures may prove too rigid for richer semantic spaces with entangled features. While the architecture implicitly encourages disentanglement, it lacks explicit alignment between latent dimensions and known factors, resulting in merely qualitative interpretability without formal semantic grounding.